# On the symmetries of the synchronization problem in Cryo-EM: Multi-Frequency Vector Diffusion Maps on the Projective Plane

**Gabriele Cesa**
Qualcomm AI Research, Amsterdam*
AMLab, University of Amsterdam
`gcesa@qti.qualcomm.com`

**Arash Behboodi**
Qualcomm AI Research, Amsterdam*
`behboodi@qti.qualcomm.com`

**Taco Cohen**
Qualcomm AI Research, Amsterdam*
`tacos@qti.qualcomm.com`

**Max Welling**
AMLab, University of Amsterdam
`welling.max@gmail.com`

## Abstract

Cryo-Electron Microscopy (Cryo-EM) is an important imaging method which allows high-resolution reconstruction of the 3D structures of biomolecules. It produces highly noisy 2D images by projecting a molecule's 3D density from random viewing directions. Because the projection directions are *unknown*, estimating the images' poses is necessary to perform the reconstruction. We focus on this task and study it under the *group synchronization* framework: if the relative poses of pairs of images can be approximated from the data, an estimation of the images' poses is given by the assignment which is most consistent with the relative ones. In particular, by studying the symmetries of cryo-EM, we show that relative poses in the group $O(2)$ provide sufficient constraints to identify the images' poses, up to the molecule's chirality. With this in mind, we improve the existing multi-frequency vector diffusion maps (MFVDM) method: by using $O(2)$ relative poses, our method not only predicts the similarity between the images' viewing directions but also recovers their poses. Hence, we can leverage all input images in a 3D reconstruction algorithm by initializing the poses with our estimation rather than just clustering and averaging the input images. We validate the recovery capabilities and robustness of our method on randomly generated synchronization graphs and a synthetic cryo-EM dataset.

## 1 Introduction

Cryo-Electron Microscopy (Cryo-EM) revolutionized the field of structural biology, enabling the reconstruction of protein structures at unprecedented resolutions. Indeed, the 2017 Nobel Prize in Chemistry was awarded to three scientists for their pioneering works on it [1]. In single-particle cryo-EM, a purified solution containing the molecule of interest is frozen on a thin film and then bombarded with electrons to obtain a 2D tomographic (integral) projection of it. The resulting image contains the projection of each copy of the molecule in the solution; in a *particle picking* phase, these projections are cropped to obtain a dataset of 2D images. Since each copy in the solution is randomly rotated in 3D, each image is a projection of the molecule's density in a random *unknown* pose. Moreover, the produced images are characterized by very low signal-to-noise ratios (SNR) [2],

---

*Qualcomm AI Research is an initiative of Qualcomm Technologies, Inc.

36th Conference on Neural Information Processing Systems (NeurIPS 2022).

e.g. see Fig. 1. The objective is reconstructing the molecule's 3D structure from these observations, but the high noise and the unknown poses make this inverse problem particularly challenging.

Common methods attempt to estimate the images' poses while performing the reconstruction. This is typically done via iterative refinement with an Expectation-Maximization (EM) algorithm [3, 4]. Unfortunately, EM-like methods are known to suffer from convergence issues due to local minima and generally require sufficiently good initializations and restarts [5]. The 3D density of known molecules can be used as initial guess but this often causes *model bias* [6]. *Ab-initio* methods, which don't rely on an initial guess, are particularly useful to initialize the iterative methods above and, therefore, are a fundamental element of the reconstruction pipeline. Moreover, a good initialization of the poses can speed up iterative refinement by providing a non-flat prior over poses, and so, reducing the search space.

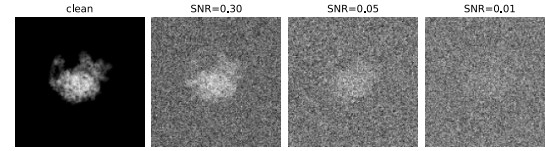

Figure 1: Example of Cryo-EM images at different SNR values.

Due to the high level of noise, a first phase of *2D classification and averaging* is performed, where a smaller denoised set of images is generated by clustering images with a similar viewing direction and averaging images within each cluster. Because this blurs out the fine details, it can reduce the final reconstruction's resolution and, therefore, it is desirable to minimize the averaging performed [7, 5]. Vector Diffusion Maps (VDM), and their variants, [8, 9, 10] are powerful methods to estimate the similarity between images' viewing directions and, therefore, provide a robust metric for clustering. In short, these methods leverage the *equivariance* of Cryo-EM to planar rotations (SO(2)) and interpret the planar rotations aligning pairs of images as an approximation of the local parallel transport over a sphere of viewing directions. The closer the viewing directions of two images are, the more consistent is the transport along different paths between them. This notion of consistency is encoded in the top eigenvectors of an Hermitian matrix representing the discretized parallel transport operator.

**Contributions and Outline**   In this work, we consider the problem of estimating Cryo-EM images' poses under the *group synchronization* framework, i.e. using the relative poses of pairs of images to recover an estimation of the absolute poses. First, in Sec. 2, we relate the symmetries of Cryo-EM with those of the synchronization task: we show that SO(2) *relative poses are not sufficient* to solve the multi-view synchronization problem, *but* O(2) *relative poses are*. Second, in Sec. 3, we use this insight to construct an improved VDM method, which is able to recover not only a similarity metric between the images, but also an estimation of each image's pose. In particular, we extend VDM by *i)* using vector diffusion over a *projective plane* (rather than a sphere) to reconstruct denoised similarity and parallel transport and by *ii)* including a *final synchronization* step to recover the absolute poses. This minor modification of VDM allows one to replace the 2D classification phase with a pose estimation phase and, therefore, leverage all the raw cryo-EM images in the final 3D reconstruction without the loss of resolution caused by averaging. Additionally, our improved VDM method shares the same spectral properties of the original VDM method and, hence, the same noise stability and recovery guarantees. We discuss the complexity of our method in Sec. 3.3 and its limitations in Sec. 3.4. Finally, we validate experimentally our theoretical results in Sec. 5.

## 1.1   The mathematics of Cryo-EM

Following [11], we consider the problem of single-particle homogeneous reconstruction and assume no image translations for simplicity. Then, a molecule's 3D density is a function $\Psi : \mathbb{R}^3 \to \mathbb{R} \in L^2(\mathbb{R}^3)$ with compact support around the origin of $\mathbb{R}^3$. An observation is a gray-scale image generated by the *integral projection* along the $Z$ axis $\Pi : L^2(\mathbb{R}^3) \to L^2(\mathbb{R}^2)$, which is defined as:

$$\Pi(\Psi)(x, y) = \int_z \Psi(x, y, z) \mathrm{d}z \quad x, y, z \in \mathbb{R} .$$

Let $\{o_i\}_{i=1}^N$ be the collection of images observed. The image $o_i := \Pi(g_i^{-1}.\Psi) \in L^2(\mathbb{R}^2)$ is the projection of a copy of the molecule $\Psi$ rotated by a random $g_i^{-1} \in \mathrm{SO}(3)$, where the action of SO(3) on $L^2(\mathbb{R}^3)$ is the standard action:

$$[g.\Psi](\boldsymbol{x}) := \Psi(g^{-1}.\boldsymbol{x}) \quad \forall g \in \mathrm{SO}(3) .$$

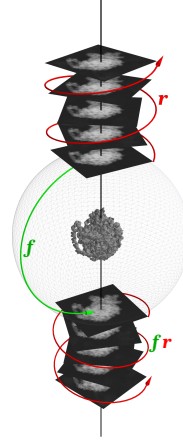

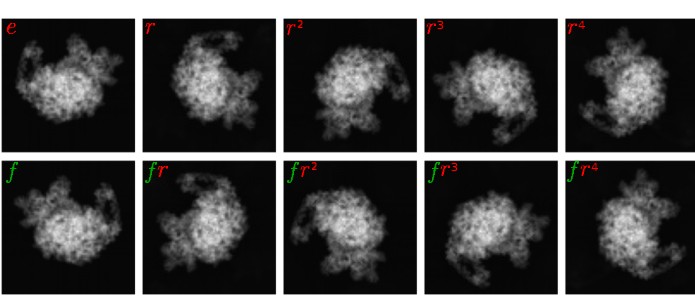

Figure 2: Generation process of Cryo-EM images. Projections from similar or opposite directions are related by elements of O(2), see also Fig. 3.

Figure 3: *First row*: images sharing a similar viewing direction with the top-left image. *Second row*: images with opposite viewing direction with respect to the top-left image. The colored group elements are the relative transformation with respect the top-left image; $e$ is the identity, $r$ is a rotation by $\frac{2\pi}{5}$ around the viewing axis $Z$ and $f$ mirrors along the $X$ axis.

An element $g_i \in \mathrm{SO}(3)$ is identified with a real orthonormal matrix $(\boldsymbol{x}_i, \boldsymbol{y}_i, \boldsymbol{z}_i) \in \mathbb{R}^{3 \times 3}$ with positive determinant. The three columns $\boldsymbol{x}_i, \boldsymbol{y}_i, \boldsymbol{z}_i \in \mathbb{R}^3$ form an orthonormal basis for $\mathbb{R}^3$. We can write:

$$o_i(x, y) = \left[ \Pi(g_i^{-1}\Psi) \right](x, y) = \int_z \Psi(x\boldsymbol{x}_i + y\boldsymbol{y}_i + z\boldsymbol{z}_i)\mathrm{d}z . \tag{1}$$

In particular, if $g_i = (\boldsymbol{x}_i, \boldsymbol{y}_i, \boldsymbol{z}_i)$, then $\boldsymbol{z}_i \in \mathbb{R}^3$ is the viewing direction along which $\Psi$ is projected, while $(\boldsymbol{x}_i, \boldsymbol{y}_i)$ defines the camera rotation around $\boldsymbol{z}_i$. We define the *projection map* $\pi(g_i) = \boldsymbol{z}_i \in \mathbb{R}^3$.

## 2 On the symmetries of the Cryo-EM synchronization problem

One of the main challenges in cryo-EM is the fact that the pose $g_i \in \mathrm{SO}(3)$ of each observation $o_i$ is unknown. This prevents one to glue the observations together to reconstruct the density $\Psi$. However, the set of observations $\{o_i\}_i$ can be used to estimate the relative poses $\{\widehat{g}_{ij} \approx g_{ij} = g_j^{-1}g_i\}_{ij}$. A synchronization problem is defined by a "synchronization graph" $\mathcal{G} = (\mathcal{V}, \mathcal{E})$, i.e. a graph with a node in $\mathcal{V}$ per image and an edge $(i, j) \in \mathcal{E}$ associated with $\widehat{g}_{ij}$ if its estimation is available. Given a synchronization graph $\mathcal{G}$, the goal is now to "synchronize" them, i.e. find a global assignment of the poses $\{\widehat{g}_i\}_i$ consistent with the estimated relative poses:

$$\widehat{g}_j \approx \widehat{g}_i \widehat{g}_{ij}^{-1} \quad \forall (i, j) \in \mathcal{G} . \tag{2}$$

**Global** $\mathrm{SO}(3)$ **ambiguity**  If $\{\widehat{g}_i\}_i$ is a solution, so is $\{g\widehat{g}_i\}_i$ for any $g \in \mathrm{SO}(3)$. This ambiguity is not unexpected and corresponds to the fact that any rotated molecule $g^{-1}.\Psi$ is an equally valid reconstruction, i.e. cryo-EM can not recover the orientation of a molecule.

### 2.1 Relative poses in $\mathrm{SO}(2)$: the impossibility of synchronization

Before solving the synchronization problem, it is necessary to estimate the relative poses $\{g_{ij} = g_j^{-1}g_i\}_{ij}$. Unfortunately, since the projection $\Pi$ loses information about the 3D structure, it is not possible to estimate the relative pose of any pair of images. However, when the viewing directions $\pi(g_i) = \boldsymbol{z}_i$ and $\pi(g_j) = \boldsymbol{z}_j$ of two images are sufficiently close, they generally differ only by a planar rotation $r_{ij} \in \mathrm{SO}(2)$ which can be directly estimated as

$$r_{ij} = \underset{r \in \mathrm{SO}(2)}{\arg\min} \|o_j - r.o_i\|_2^2 \text{ if } \pi(g_i) \approx \pi(g_j) . \tag{3}$$

where $r$ acts on image $o_i$ by rotating it. The closer $\boldsymbol{z}_i$ and $\boldsymbol{z}_j$ are, the better $r_{ij}$ approximates $g_{ij}$. Indeed, if $\boldsymbol{z}_i = \boldsymbol{z}_j$, then $g_{ij} = g_j^{-1}g_i$ has form[2] $\begin{bmatrix} * & * & 0 \\ * & * & 0 \\ 0 & 0 & 1 \end{bmatrix}$, where the top-left $2 \times 2$ block is an $\mathrm{SO}(2)$

---

[2]This is shown using the orthogonality of $g_i$ and $g_j$'s columns.

matrix. We identify all the rotations of this form as the subgroup[3] $\mathrm{SO}(2) \cong R_z < \mathrm{SO}(3)$ of planar rotations around the $Z$ axis. The projection operator $\Pi$ is $\mathrm{SO}(2)$-**equivariant**[4], i.e.:

$$\Pi(r.\Psi) = r.\Pi(\Psi) \quad \forall r \in R_z \cong \mathrm{SO}(2) . \tag{4}$$

Hence, if $g_j^{-1} = r g_i^{-1}$, it follows that $o_j = r.o_i$. For example, all projections generated by the top view in Fig 2 are related by a rotations $r$ as shown in the top row of Fig. 3.

**Global** $\mathrm{SO}(2)$ **ambiguity** While this is a common approach in the literature, e.g. in VDM [8, 12, 10, 13], it does not provide sufficient information to estimate the poses $\{g_i\}_i$. Indeed, the estimated relative rotations $\{r_{ij} \in \mathrm{SO}(2)\}_{(ij) \in \mathcal{G}}$ yield the following constraints:

$$g_j \approx g_i r_{ij}^{-1} \quad \forall(i,j) \in \mathcal{G} .$$

Since $\mathrm{SO}(2)$ is *abelian*, if $\{\widehat{g}_i\}_i$ is a solution to this set of constraints, then $\{\widehat{g}_i r\}_i$ is an equally valid solution for any $r \in R_z \cong \mathrm{SO}(2)$:

$$\widehat{g}_j \approx \widehat{g}_i r_{ij}^{-1} \implies \widehat{g}_j r \approx \widehat{g}_i r_{ij}^{-1} r \implies \widehat{g}_j r \approx (\widehat{g}_i r)\, r_{ij}^{-1}$$

This implies that any method which estimates the poses $\{g_i\}_i$ from the estimated relative ones $\{r_{ij} \in \mathrm{SO}(2)\}_{ij}$ will only be able to recover them up to a global rotation $r \in R_z$. This global ambiguity prevents using these estimations to directly invert the linear projection $\Pi$ and recover the original molecule $\Psi$, since it would result in averaging randomly rotated versions of the images.

Nevertheless, the ambiguous poses $\{g_i r\}_i$ still contain information about the viewing direction of the projections. Indeed, note that $\pi(g_i r) = \pi(g_i)$, i.e. the viewing directions are invariant to this global ambiguity and, therefore, can be recovered from this method. Previous works such as VDM [8, 12, 10] exploit this invariance to estimate the relative distance between different images. This distance is generally used for *class averaging* i.e. to cluster images and average images within each cluster to generate a smaller, de-noised, images dataset.

## 2.2 Relative poses in $\mathrm{O}(2)$: a sufficient condition for synchronization

In this work, we solve this global $\mathrm{SO}(2)$ ambiguity and directly estimate the absolute poses $\{g_i\}_i$ of the images. To do so, we will exploit another symmetry of Cryo-EM. Indeed, the tomographic projection $\Pi$ is also invariant to *mirroring* along the Z axis $\mathrm{m}_z = \begin{bmatrix} 1 & 0 & 0 \\ 0 & 1 & 0 \\ 0 & 0 & -1 \end{bmatrix} \in \mathrm{O}(3)$ i.e.:

$$\Pi(\mathrm{m}_z .\Psi) = \int_z \Psi(x,y,-z)dz = \int_z \Psi(x,y,z)dz = \Pi(\Psi)$$

This additional symmetry turns out useful to break the global $\mathrm{SO}(2)$ symmetry explained earlier.

First, let $r_y = \begin{bmatrix} -1 & 0 & 0 \\ 0 & 1 & 0 \\ 0 & 0 & -1 \end{bmatrix} \in \mathrm{SO}(3)$ be a $\pi$ rotation along the $Y$ axis; note that $r_y = \mathrm{m}_z \mathrm{m}_x$, where $\mathrm{m}_x$ is a mirroring along the $X$ axis. Then, let $\mathrm{f} = \begin{bmatrix} -1 & 0 \\ 0 & 1 \end{bmatrix} \in \mathrm{O}(2)$ be the flip of a planar image along the $X$ axis. Then:

$$\Pi(r_y.\Psi) = \Pi(\mathrm{m}_z \mathrm{m}_x .\Psi) = \Pi(\mathrm{m}_x .\Psi) = \mathrm{f} .\Pi(\Psi) , \tag{5}$$

i.e. the projection operator $\Pi$ is **flip equivariant**: projections along a direction ($\Pi(g_i^{-1}.\Psi)$) are related to projections from the opposite direction ($\mathrm{f} .\Pi(r_y g_i^{-1}.\Psi)$) by a planar reflection $\mathrm{f}$. For example, all projections generated by the bottom view in Fig 2 are related by a rotation $r$ followed by the reflection $\mathrm{f}$ with the images generated from the top view, as shown in the bottom row of Fig. 3. To simplify the notation, we will also let $r_y$ act on a image; this action should be intended as that of $\mathrm{f}$.

That means that we can not only estimate the relative pose $g_{ij} \approx h_{ij} \in R_z$ when the viewing directions $\pi(g_i)$ and $\pi(g_j)$ are sufficiently close, but also $h_{ij} \in H \cong \mathrm{O}(2)$ when $\pi(g_i) \approx -\pi(g_j)$, i.e.:

$$h_{ij} = \underset{h \in H \cong \mathrm{O}(2)}{\arg\min} \|o_j - h.o_i\|_2^2 \quad \text{if } \pi(g_i) \approx \pm\pi(g_j) . \tag{6}$$

where $H \cong \mathrm{O}(2)$ is the subgroup[5] of $\mathrm{SO}(3)$ containing $R_z$ (planar rotations along the $Z$ axis) and $r_y$.

---

[3]$\mathrm{SO}(3)$ has many subgroups isomorphic to $\mathrm{SO}(2)$ so we use $R_z$ to indicate this particular one.

[4]For simplicity, the action of $R_z$ on 3D densities or 2D images is not distinguished.

[5]More precisely, $H := R_z \rtimes \{e, r_y\} \cong \mathrm{O}(2)$.

Table 1: Summary of the Cryo-EM symmetries.

| | description of the transformation | symmetry name |
|---|---|---|
| $o_i = \Pi(g_i^{-1}\Psi)$ | $g_i \in \mathrm{SO}(3)$ | generative process |
| $r.o_i = \Pi(rg_i^{-1}\Psi)$ | $r \in \mathrm{SO}(2)$ any rotation around $Z$ axis | $\mathrm{SO}(2)$ equivariance |
| $o_i = \Pi(\mathrm{m}_z\, g_i^{-1}\Psi)$ | $\mathrm{m}_z \in \mathrm{O}(3)$ mirroring along $Z$ axis | Mirroring ($Z$) invariance |
| $\mathrm{f}.o_i = \Pi(m_x g_i^{-1}\Psi)$ | $\mathrm{m}_x \in \mathrm{O}(3), \mathrm{f} \in \mathrm{O}(2)$ mirroring along $X$ axis | Mirroring ($XY$) equivariance |
| $\mathrm{f}.o_i = \Pi(r_y g_i^{-1}\Psi)$ | $r_y = \mathrm{m}_x\, \mathrm{m}_z \in \mathrm{SO}(3)$ rotation by $\pi$ around $Y$ axis | Flip equivariance |

**Global chirality ambiguity**  The relative poses $\{h_{ij} \in \mathrm{O}(2)\}_{ij}$ still don't fully constrain the absolute ones; if $\{\widehat{g}_i\}_i$ is a solution, so is $\{\widehat{g}_i r_z\}_i$, with $r_z \in R_z$ being a $\pi$-rotation around the $Z$ axis[6]:

$$\widehat{g}_j \approx \widehat{g}_i h_{ij}^{-1} \implies \widehat{g}_j r_z \approx \widehat{g}_i h_{ij}^{-1} r_z \implies \widehat{g}_j r_z \approx (\widehat{g}_i r_z)\, h_{ij}^{-1}$$

This ambiguity is related to the well known problem that Cryo-EM can not recover the **chirality** of a molecule. To see why, let $\mathrm{i} = \begin{bmatrix} -1 & 0 & 0 \\ 0 & -1 & 0 \\ 0 & 0 & -1 \end{bmatrix}$ be the *inversion*, i.e. a mirroring along all axes; note that $\mathrm{m}_z = r_z\, \mathrm{i}$ and that $\mathrm{i}$ commutes with any element $g \in \mathrm{SO}(3)$. Then:

$$o_i = \Pi(g_i^{-1}.\Psi) = \Pi(\mathrm{m}_z\, g_i^{-1}.\Psi) = \Pi(r_z\, \mathrm{i}\, g_i^{-1}.\Psi) = \Pi(r_z g_i^{-1}.(\mathrm{i}\,.\Psi)) = \Pi((g_i r_z)^{-1}.(\mathrm{i}\,.\Psi))$$

i.e., $o_i$ is equally likely to be generated by the molecule $\Psi$ with pose $g_i$ or its mirrored version $\mathrm{i}\,.\Psi$ with pose $g_i r_z$. Together with the global $\mathrm{SO}(3)$ symmetry introduced earlier, this means Cryo-EM intrinsically suffers from a **global $\mathrm{O}(3)$ ambiguity** which we can not hope to resolve. Nevertheless, since $r_z$ is the *only* element of $\mathrm{SO}(3)$ commuting with all elements in $H \cong \mathrm{O}(2)$, we have that relative poses in $H$ constrain the synchronization problem precisely up to this $\mathrm{O}(3)$ symmetry and, thus provide *sufficient* constraints to identify the images' poses. We provide a theoretical argument based on Vector Diffusion Maps (VDM) [9] supporting this *informal* claim in Apx. C.

## 3 Multi-Frequency Vector Diffusion Maps on the Projective Plane

In this section, we use the new insights on the symmetries of cryo-EM to enable the Vector Diffusion Map (VDM) method to recover the poses $\{g_i\}_i$. The key component is the Graph Connection Laplacian (GCL) of the synchronization graph $\mathcal{G}$ with the estimated relative poses $\{h_{ij} \in \mathrm{O}(2)\}_{ij}$ on its edges. Under certain conditions, its eigenvectors converge to the eigen-vector-fields of the connection Laplacian operator on the projective plane, which contain information about the absolute poses. Our method involves *two steps*: a *denoising step* (like the original VDM) and a *synchronization step*; Alg. 1 provides an overview of the proposed method. We study its computational complexity in Sec. 3.3.

### 3.1 Quotient Spaces and Projective Plane

Each image $o_i$ is uniquely identified by an element $g_i \in \mathrm{SO}(3)$ via Eq. 1, assuming the molecule has no symmetries. Two images $o_i$ and $o_j$ related by a rotation $r_{ij} \in R_z$ contain the same information and are considered equivalent. We can define a similar equivalence relation on their respective poses, i.e. $g_i \sim g_j = g_i r_{ij}$. This defines the quotient space $\mathrm{SO}(3)/R_z$ which is isomorphic to the *sphere* $\mathcal{S}^2$. An element $g_i \in \mathrm{SO}(3)$ is interpreted as a point $\pi(g_i) = z_i$ in $\mathcal{S}^2$ and a choice of *frame* on it, i.e. the choice of basis[7] $(x_i, y_i)$ for the tangent space at $z_i$ or, equivalently, the rotation of the camera around the axis $z_i$. Then, the relative rotation $r_{ij} \in R_z$ estimated in Eq. 3 is interpreted as the unique rotation such that $g_j r_{ij}^{-1}$ is the *parallel transport* of $g_i$ along the shortest geodesic connecting $\pi(g_i)$ to $\pi(g_j)$ (this is unique unless $\pi(g_i) = -\pi(g_j)$). Here, we want to leverage the larger symmetry $H$. We define the new equivalence relation $g \sim gh$ for all $h \in H$ on $\mathrm{SO}(3)$, which determines the quotient space $\mathrm{SO}(3)/H$, isomorphic to the *projective plane*[8] $\mathrm{P}\mathbb{R}^2$. While it is harder to visualize, elements of $\mathrm{SO}(3)$ are now interpreted as points in $\mathrm{P}\mathbb{R}^2$ together with a choice of frame for their tangent spaces[9]. Finally, the relative pose $h_{ij} \in H$ from Eq. 6 is interpreted as the element such that $g_j h_{ij}^{-1}$ is the *parallel transport* of $g_i$ along the shortest geodesic between the projections of $g_i$ and $g_j$ on $\mathrm{P}\mathbb{R}^2$.

---

[6] One can verify that $r_z$ commutes with any $r \in R_z$ and $r_x$.

[7] Indeed, $(x_i, y_i)$ span a 2D plane through the origin parallel to the tangent plane at $z_i$.

[8] The projective plane can be thought as a sphere where antipodal points are identified together.

[9] By moving from $\mathcal{S}^2$ to $\mathrm{P}\mathbb{R}^2$, we change the *structure group* and consider frames in $\mathrm{O}(2)$ rather than $\mathrm{SO}(2)$.

## 3.2 Vector Diffusion Maps on $\mathbb{PR}^2$

An initial synchronization graph $\mathcal{G}$ can be generated by comparing all pairs of images using Eq. 6 and preserving only the edges with smallest distances. Ideally, this graph is a discretization of $\mathbb{PR}^2 \cong \mathrm{SO}(3)/H$ with local connectivity, but the noise on the images negatively affects the estimated distances and relative poses, introducing "shortcut" edges in this graph. Therefore, the first step of our method is re-estimating the geodesic distances and the relative poses of the points in the graph.

Like previous works, we leverage the consistency of the local alignments to do this. Indeed, since a manifold is locally Euclidean, parallel transport along sufficiently short paths is approximately path-independent in the local neighborhood of a point. On the other hand, the further away two points are, the more inconsistent the cycles through them will be. When summing the transport over multiple paths between two points, inconsistent paths tend to average out.

We use the synchronization graph to build a discretization of this local parallel transport operator. Let the graph connection Laplacian $W$ be the $2N \times 2N$ matrix with its $i,j$-th $2 \times 2$ block defined as:

$$W(i,j) = \begin{cases} \rho(h_{ij}) & \text{if } (i,j) \in \mathcal{G} \\ 0 & \text{otherwise} \end{cases} \in \mathbb{R}^{2 \times 2} \tag{7}$$

where $\rho(h_{ij})$ is the standard representation of $h_{ij} \in \mathrm{O}(2)$ as a $2 \times 2$ orthogonal matrix. Let $D$ be the $2N \times 2N$ diagonal matrix containing the *degree* of each node in $\mathcal{G}$ (repeated twice) in its diagonal. The matrix $W$ acts by parallel-transporting tangent vectors along each edge of the graph while $A := D^{-1}W$ averages all vectors transported to a node. Then, $W^t$ transports vectors along all length-$t$ paths in $\mathcal{G}$ and the block $A^t(i,j)$ is the average of the transport over all length-$t$ paths from $i$ to $j$. If the paths are mostly inconsistent, $A^t(i,j)$ tends to $0$, being the average of uncorrelated orthogonal matrices. Hence, $A^t(i,j)$ provides information about the geodesic distance between $i$ and $j$.

Like [12], we take a **multi-frequency** approach to improve robustness to noise. In practice, we build a different graph connection Laplacian $A_k$ for different choices[10] of *irreducible representation* (irrep) $\rho_k$ of $\mathrm{O}(2)$. For $k > 0$, $\rho_k$ is a $2 \times 2$ orthogonal matrix with rotational frequency $k$; see Apx. A.

In the limit $t \mapsto \infty$, the power iteration of $A_k^t$ converges to its top eigenspace and $A_k^t(i,j)$ will be the product of the top eigenvectors of $A_k$. It is therefore important to discuss the spectral properties of $A_k$. [14] proved that the normalized graph connection Laplacian is a discretization of the connection Laplacian operator and that, under certain conditions, its eigenvectors and eigenvalues converge to those of the connection Laplacian operator. In Apx. B, we study the spectral properties of the connection Laplacian operator over $\mathbb{PR}^2$ and relate it with the operator over $\mathcal{S}^2$ studied in [13]. In particular, we show that the eigenspaces of our Laplacian operator over $\mathbb{PR}^2$ are subspaces of the eigenspaces of the spherical Laplacian and they share the same eigenvalues. This also implies our graph connection Laplacians enjoy the same spectral properties studied in [13], including the spectral gap and the robustness to noise.

For frequency $k > 0$, the top eigen-space of $A_k$ is $2k+1$ dimensional. Additionally, denote $\varphi_k : \mathcal{V} \to \mathbb{R}^{(2k+1) \times 2}$ the stack of the top $2k+1$ eigen-vectors. Let $\underline{\varphi_k}(i) = \frac{\sqrt{2}}{\|\varphi_k(i)\|_F} \varphi_k(i)$ and define $\widehat{A_k}(i,j) = \underline{\varphi_k}(j)^T \underline{\varphi_k}(i) \in \mathbb{R}^{2 \times 2}$. Then:

**Theorem 3.1.** *Let* $s_{ij} = \langle \mathbf{z}_j, \mathbf{z}_i \rangle = \langle \pi(g_i), \pi(g_j) \rangle \in [-1,1]$ *be the* cosine similarity *of the viewing directions* $\pi(g_i), \pi(g_j) \in \mathcal{S}^2$. *The two following identities hold:*

$$\frac{1}{2} \left\| (\widehat{A_k})(i,j) \right\|_F^2 \approx \left( \frac{1+s_{ij}}{2} \right)^{2k} + \left( \frac{1-s_{ij}}{2} \right)^{2k} \quad , \quad \det(\widehat{A_k}(i,j)) \approx \left( \frac{1+s_{ij}}{2} \right)^{2k} - \left( \frac{1-s_{ij}}{2} \right)^{2k}$$

The proof is in Apx. D. Since we need a similarity over $\mathbb{PR}^2$ rather than $\mathcal{S}^2$, we are interested in $w_{ij} := |s_{ij}|$. In other words, the top eigenspace of $A_k$ allows the estimation of the geodesic distance between points. In practice, $d_k > 2k+1$ top-eigenvectors can be used, as in [8, 12].

**Denoised similarity and parallel transport** The information obtained from the different $A_k$ matrices can be combined in multiple ways to obtain an estimation of $s_{ij}$. Define

$$S_{ij}^{k\pm} := \frac{1}{4} \left\| (\widehat{A_k})(i,j) \right\|_F^2 \pm \frac{1}{2} \det(\widehat{A_k}(i,j)) \approx \left( \frac{1 \pm s_{ij}}{2} \right)^{2k} \tag{8}$$

---

[10]The standard representation of $\mathrm{O}(2)$ as rotation matrices correspond to $k = 1$.

**Algorithm 1** MFVDM based synchronization over $\mathrm{P}\mathbb{R}^2$

---

**Require:** $\{o_i \mid i = 1, \ldots, N \in \mathcal{V}\}$ the input raw images, max frequency $L > 0$

1: $\mathcal{G}(\mathcal{V}, \mathcal{E}) \leftarrow$ synchronization-graph $\left(\{h_{ij} = \arg\min_h \|o_j - h.o_i\|_2^2\}_{ij}, \{d_{ij} = \min_h \|o_j - h.o_i\|_2^2\}_{ij}\right)$

2: **for** $0 < k \leq L$ **do**

3:      $A_k \leftarrow [\rho_k(h_{ij}) \text{ if } (i,j) \in \mathcal{E} \text{ else } 0]_{ij}$            ▷ Frequency-$k$ graph connection Laplacian; Eq. 7

4:      $\varphi_k \leftarrow$ top-eigenvectors$(A_k, 2k+1)$

5:      $\widehat{A_k} \leftarrow \left[ \left( \frac{\sqrt{2}}{\|\varphi_k(j)\|_F} \varphi_k(j) \right)^T \left( \frac{\sqrt{2}}{\|\varphi_k(i)\|_F} \varphi_k(i) \right) \right]_{ij}$

6:      $S_{ij}^{k\pm} \leftarrow \frac{1}{4} \left\| (\widehat{A_k})(i,j) \right\|_F^2 \pm \frac{1}{2} \det(\widehat{A_k}(i,j)) \quad \forall i,j \in \mathcal{V}$            ▷ Eq. 8

7: $\widehat{s}_{ij}^{\pm} \leftarrow \exp\left( \frac{1}{L} \sum_{k=1}^{L} \frac{\log S_{ij}^{k\pm}}{2k} \right)$

8: $\widehat{s}_{ij} \leftarrow \mathrm{sign}(\widehat{s}_{ij}^+ - \widehat{s}_{ij}^-) \cdot \max(\widehat{s}_{ij}^+, \widehat{s}_{ij}^-)$            ▷ **Denoised similarity** over $\mathrm{P}\mathbb{R}^2$; Eq. 10

9: $\widehat{h}_{ij} \leftarrow \arg\max_h \sum_{0<k\leq L} \sqrt{\dim_{\rho_k}} \,\mathrm{Tr}\left( \widehat{A_k}(i,j)\rho_k(h)^T \right)$            ▷ **Denoised parallel transport**; Eq. 11

10: $\mathcal{G}'(\mathcal{V}, \mathcal{E}') \leftarrow$ synchronization-graph $\left( \{\widehat{h}_{ij}\}_{ij}, \{|\widehat{s}_{ij}|\}_{ij} \right)$            ▷ Build denoised Synchronization Graph

11: $A' \leftarrow \left[ \rho_1(\widehat{h}_{ij}) \text{ if } (i,j) \in \mathcal{E}' \text{ else } 0 \right]_{ij}$            ▷ Build denoised graph connection Laplacian; Eq. 7

12: $x, y \leftarrow \varphi \leftarrow$ top-eigenvectors$(A', 3)$            ▷ **Final Synchronization**

13: $\widehat{g}_i \leftarrow$ projectSVD$(x(i), y(i), x(i) \times y(i))$

---

Then, if $L$ is the largest frequency considered, the similarity $s_{ij}$ can be estimated as

$$\tilde{s}_{ij} = \widehat{s}_{ij}^+ - \widehat{s}_{ij}^- \quad \text{with} \quad \widehat{s}_{ij}^{\pm} = \exp\left( \frac{1}{L} \sum_{k=1}^{L} \frac{\log S_{ij}^{k\pm}}{2k} \right). \tag{9}$$

In practice, we found the following estimator more robust for the nearest neighbors search:

$$\widehat{s}_{ij} = \mathrm{sign}(\widehat{s}_{ij}^+ - \widehat{s}_{ij}^-) \cdot \max(\widehat{s}_{ij}^+, \widehat{s}_{ij}^-), \quad \text{and} \quad \widehat{w}_{ij} = |\widehat{s}_{ij}| = \max(\widehat{s}_{ij}^+, \widehat{s}_{ij}^-) \tag{10}$$

although it approximates $\frac{1 \pm s_{ij}}{2}$, rather than $s_{ij}$. In Apx. E, we discuss an alternative estimator which can be expressed as a dot product, enabling the use of a fast $K$-nearest neighbors ($K$-NN) search rather than computing the similarity between all pairs. Additionally, note that, due to the decaying spectrum of the Laplacian operators, the noise has a stronger effect on the lowest eigenvalues; discarding the lower eigenvectors also helps denoising the GCL matrices. That means that the top eigenvectors of $A_k$ can be used to partially denoise the frequency-$k$ parallel transport. Next, we combine all denoised blocks $\{\widehat{A_k}(i,j)\}_k$ by interpreting them as the Fourier coefficients of a function on $H$; we estimate the relative pose with the element $h \in H$ maximizing this function:

$$\widehat{h}_{ij} = \arg\max_{h \in H} \sum_{\rho_k \in \widehat{H}} \sqrt{\dim_{\rho_k}} \,\mathrm{Tr}\left( \widehat{A_k}(i,j)\rho_k(h)^T \right) \tag{11}$$

**Final Synchronization** So far, we have used the MFVDM method on $\mathrm{P}\mathbb{R}^2$ to estimate the similarity $\widehat{w}_{ij} = |\widehat{s}_{ij}|$ and parallel transport $\widehat{h}_{ij}$ over $\mathrm{P}\mathbb{R}^2$. With respect to [12], our method includes a final synchronization step, where these quantities are used to recover $\{\widehat{g}_i\}_i$. We construct a new graph connection Laplacian matrix $A'$ as in Eq. 7 using the new estimations and the standard representation ($k = 1$) of $\mathrm{O}(2)$. The top eigen-space of $A'$ is 3 dimensional and its 3 top eigenvectors define a tangent frame at each node in the following way. Let $\varphi : \mathcal{V} \to \mathbb{R}^{3 \times 2}$ be the stack of the top 3 eigen-vectors, $\underline{\varphi}(i) = \frac{\sqrt{2}}{\|\varphi(i)\|_F} \varphi(i)$ and $x : \mathcal{V} \to \mathbb{R}^3$ and $y : \mathcal{V} \to \mathbb{R}^3$ be its two columns. Then, $x(i)$ and $y(i) \in \mathbb{R}^3$ define a basis for the tangent space at $\pi(g_i) \in \mathcal{S}^2$, see Apx. C. Recall that $g_i = (\boldsymbol{x}_i, \boldsymbol{y}_i, \boldsymbol{z}_i) \in \mathrm{SO}(3)$, then $x(i)$ and $y(i)$ approximates respectively $\boldsymbol{x}_i$ and $\boldsymbol{y}_i$, while $\boldsymbol{z}_i$ can be recovered[11] as $z(i) = x(i) \times y(i)$. Note that recovered poses $\{\widehat{g}_i = (x(i), y(i), z(i))\}_i$ present the global $\mathrm{O}(3)$ symmetry discussed in Sec. 2. Indeed, eigenvalue decomposition is unique up to an orthogonal change of basis in each eigenspace. Since the top eigenspace is 3 dimensional, the solution is unique up to a global $g \in \mathrm{O}(3)$ transformation, i.e. if $\varphi$ is a set of orthogonal eigenvectors,

---

[11]$\langle z(j), z(i) \rangle = \langle x(j) \times y(j), x(i) \times y(i) \rangle = \det\left( \underline{\varphi}(j)^T \underline{\varphi}(i) \right) \approx s_{ij}$ by using Theorem 3.1 with $k = 1$.

so is $g\varphi$, defined as $[g\varphi](i) = g\varphi(i)$. Hence, all tangent frames $\{(x(i), y(i))\}_i$ can be simultaneously rotated by any $g \in \mathrm{O}(3)$. Finally, because $x(i)$ and $y(i)$ are not perfectly unitary and orthogonal to each other, the matrix $(x(i), y(i), z(i))$ will not be orthogonal; therefore, we project it to the closest $\mathrm{SO}(3)$ element via SVD.

### 3.3 Cryo-EM Pipeline and Computational Complexity

We study the complexity of our method in a pipeline similar to the one implemented in ASPIRE and described in [15]. Assume a dataset of $N$ images of resolution $D \times D$; consider the MFVDM algorithm with frequencies up to $L$ and that at most $M$ top-eigenvectors of $A_k$ are computed for each frequency $k$. The pipeline consists of the following three stages, with relative complexity:

**Preprocessing** $O(ND^3 + Nk \log N)$: a number of invariant features are built using (fast) steerable PCA $O(ND^3)$ [16] and the *bispectrum*, and are used for a $K$-nearest neighbors ($K$-NN) search $O(NK \log N)$ as in [15]. Then, the $\mathrm{O}(2)$ relative alignments of each pair are estimated in $O(ND^2 \log D + NKD^2)$ by leveraging Polar and Fast Fourier Transforms (FFT).

**MFVDM denoising** $O(NLM^2 + NKLM^2 \log N)$: the eigenvalue decomposition of the matrices $\{A_k\}_k$ can be accelerated to $O(NL(M^2 + MK)) \approx O(NLM^2)$ as in [12]. The denoised similarities $\widehat{s_{ij}}$ cost $O(N^2)$. By aggregating multiple frequencies without log as in [12], however, a faster $K$-NN search in $O(NKLM^2 \log N)$ can be used, see Apx. E. Finally, the denoised parallel transports $\widehat{h}_{ij}$ between neighbors are computed in $O(NKL \log L)$ with an FFT.

**Synchronization** $O(NK)$: since only the top 3 eigenvectors of $A'$ are required, the eigenvalue decomposition only costs $O(NK)$.

### 3.4 Limitations

The main limitation of our method is that VDM assumes uniformly distributed poses, which is not generally the case for real cryo-EM data. Renormalization techniques such as [17, 18] can help relaxing this requirement. Our analysis of the synchronization symmetries, however, is not limited to VDM but applies to any cryo-EM synchronization method. Moreover, 2D classification or pose estimation methods which do not explicitly account for the image formation model might fail at the lowest SNR regimes [5]. Indeed, if the noise is too high, the relative poses estimation itself can be impossible and increasing the number of observations can not solve this problem [19, 20], although methods like steerable PCA [16] can alleviate that. Nevertheless, in practice, 2D classification with VDM is leveraged as a denoising step in reconstruction pipelines like ASPIRE; our method can replace this step, providing pose estimation and reducing the need for averaging. Finally, while we assumed non-symmetric molecules, our method can handle symmetries, provided they are known a priori, since they enforce precise sparsity patterns in the spectrum of the diffusion operators; we leave this as future work.

## 4 Related Works

Vector-Diffusion Maps (VDM) were initially proposed in [9, 8] for solving the cryo-EM synchronization problem with $\mathrm{SO}(2)$ constraints. The $\mathrm{O}(2)$ symmetry was noted also in [8] which suggests augmenting a dataset with mirrored images; still, it only uses $\mathrm{SO}(2)$ relative alignments and, therefore, recovers only a similarity metric over the sphere. VDMs were later extended to a multi-frequency setting in [12, 10, 13]. [21] describes a strategy based on the classical *method of moments* to solve the generic problem of recovering orbits from invariants. Additionally, the authors provide theoretical analysis of sample complexity of such problems; in particular, they show that in the cryo-EM synchronization problem with uniform viewing distribution, the sample complexity scales as $\frac{1}{\mathrm{SNR}^3}$. Some works exploit the property that the Fourier transforms of any pair of cryo-EM images must agree on a line passing through the origin (*common lines approach*) to define a set of constraints on each pair of absolute poses. [22] use a Semi-Definite Programming (SDP) relaxation to find the solution; however, SDP is known for its high computational cost. Instead, [7, 23] rely on a faster spectral relaxation of the problem. Still, the estimation of common lines itself is expensive - since it requires comparing each pair of images - and very sensitive to noise; for this reason, it usually follows a first 2D classification and averaging phase [24]. Popular reconstruction methods include RELION [25] and cryoSPARC [26], which are based on an EM-like algorithm. [27] suggested using stochastic

rather than batch optimization for iterative refinement, thereby improving the computational cost. Recent advances employ neural networks optimized by SGD to represent the 3D density [28, 29]. Note that these methods still need to periodically perform an expensive search over the images' poses and [29] found resetting the model's weights throughout training essential for convergence.

## 5 Experiments

We first evaluate our method on a synchronization task over some synthetic datasets of vectors which share the same symmetries of cryo-EM. Then, we include our method in a simple 3D reconstruction pipeline which we test on synthetic cryo-EM data. We include additional experiments in Apx. F.

### 5.1 Synthetic Vector dataset

We generate a number of synthetic datasets which imitate the symmetry of the cryo-EM generative process. To do so, we first construct a matrix[12] $M \in \mathbb{R}^{d \times b}$ which has the symmetries of $\Pi$ described in Table 1, where $H$ acts on the left with a $d$-dimensional representation $\rho$ of the group $O(2)$ while $O(3)$ acts on its right with a $b$-dimensional representation $\psi$. Both representations contain irreps of different frequencies of the two groups. See Apx. A for more details about the representations of these groups. A dataset is generated by sampling a random vector $\Psi \in \mathbb{R}^b$, a set of random poses $\{g_i \in SO(3)\}_i^n$ and then evaluating $\{o_i = M\psi(g_i^{-1})\Psi + \frac{1}{\lambda}\epsilon_i \in \mathbb{R}^d\}_i$, where $\epsilon_i$'s entries are normally distributed and $\lambda^2$ is the SNR.

We evaluate variants of our method on this data and compare it with MFVDM with $SO(2)$ poses. We use the $SO(3)$ correlation[13] $\frac{1}{n\sqrt{3}} \left\| \sum_i g_i \widehat{g_i}^{-1} \right\|_F$ and the correlation between the real and estimated cosine similarities $\{s_{ij}\}_{ij}$ as metrics. Fig. 4 shows the $SO(3)$ correlation achieved by our method in different settings; in the low-noise regime, perfect reconstruction is achieved. As expected, in the highest noise regime, reconstruction is almost impossible as increasing the number of samples helps only marginally (SNR=0.16). Fig. 5 compares our method with the original MFVDM-$SO(2)$, for two choices of maximum frequency $L$. While the original estimator from MFVDM is stable when $s_{ij} \gg 0$, our estimator from Eq. 10 is stable when $|s_{ij}| \gg 0$. We also observe the effect of the multi-frequency approach: leveraging higher frequencies provide a more robust estimation.

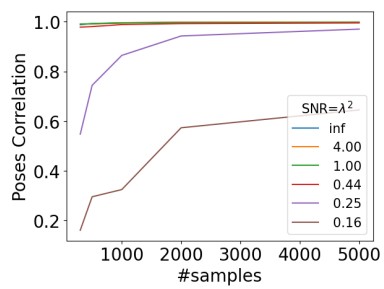

Figure 4: Correlation with the $SO(3)$ poses in different variations of the vector dataset.

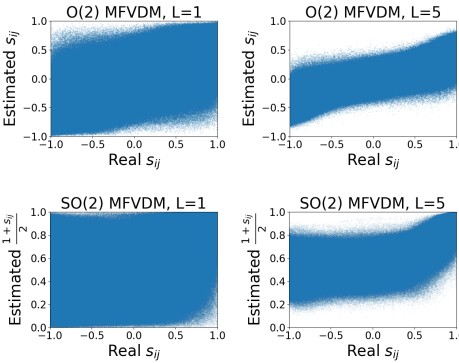

Figure 5: Estimated $\{\tilde{s}_{ij}\}_{ij}$ vs real $\{s_{ij}\}_{ij}$ cosine similarities in the synthetic vector dataset. We compare our method with $O(2)$ relative poses with the original MFVDM with $SO(2)$ poses.

### 5.2 3D reconstruction of synthetic cryo-EM data

For a more realistic evaluation, we integrate our method in the pipeline described in Sec. 3.3 and test it on the 70S Ribosome [30] structure from the Protein Data Bank database (structure 5O60). See Apx. F.2 for more details. In summary, a *first synchronization graph* is built via two $K$-NN searched by using invariant features generated with Steerable PCA and bispectrum [15] (we pick $K$ rotated and $K$ mirrored neighbors). Then, the estimated relative poses can, optionally, be *denoised* with MFVDM, either using $SO(2)$ or $O(2)$ and with different values of the maximum frequency $L$. If MFVDM-$SO(2)$ is used, information about reflections is lost, hence our VDM-based synchronization

---

[12]$M$ contains a subset of the matrix decomposing $\mathrm{Ind}_{H \times \{e, m_z\}}^{O(3)} \rho$ into $O(3)$ irreps.

[13]Note that this metric is invariant to the global $O(3)$ symmetry.

Table 2: Pose correlation (and error in radians) with different methods.

| Synchronization | | VDM (our) | VDM (our) | VDM (our) | CL | CL | CL | CL |
| --- | --- | --- | --- | --- | --- | --- | --- | --- |
| | Denoising | None | $O(2), L=5$ | $O(2), L=10$ | None | $SO(2), L=5$ | $O(2), L=5$ | $O(2), L=10$ |
| N | SNR | | | | | | | |
| 3000 | 0.35 | 99.6 (0.1) | 99.3 (0.13) | 99.3 (0.13) | | 99.6 (0.09) | 99.7 (0.08) | 99.7 (0.08) |
| 5000 | 0.12 | 82.0 (0.69) | 97.5 (0.22) | 97.5 (0.21) | 98.0 (0.20) | 97.9 (0.18) | 99.0 (0.14) | 99.0 (0.14) |
| 5000 | 0.05 | 60.1 (1.11) | 63.4 (1.05) | 63.5 (1.05) | 64.6 (1.03) | 71.4 (0.87) | 75.9 (0.77) | 76.6 (0.75) |

can not be used later. Next, the dataset can be clustered to $n < N$ images and denoised by *averaging* nearest neighbors.

As a *synchronization* baseline, we consider the *common line* method (CL) from [7, 23, 24]. Note that the estimation of common lines estimation is more sensitive to noise. Moreover, the method requires $O(N^2)$ comparisons to find common lines between each pair of images; in comparison, MFVDM only needs to estimate the relative poses of $O(NK)$ nearest neighbors. For these reasons, the CL method is usually applied on the smaller subset of $n < N$ averaged images.

Once the poses are estimated, we perform a *first 3D reconstruction* using an algorithm inspired by [31] which optimizes via SGD a Gaussian posterior over the 3D density leveraging the Fourier Slice Theorem. See also Apx. F.3 for more details about it and other experiments. Finally, this low-resolution reconstruction is used to initialize the 3D Refine method in RELION [25], which performs the final *refinement*. RELION also re-estimates the images' poses and the final resolution using the gold-standard Fourier Shell Coefficient. Since RELION is based on an EM method, it can show large variance in the final results and, sometimes, fail to converge. To reduce the impact of this on the variance of the results, for each method, we perform the initial reconstruction three times using the method above and, for each reconstruction, we consider the best run of RELION out of three trials. In the tables, we report the mean and the standard deviation over the three initial reconstructions of the corresponding best RELION trials.

In our experiments, we use images of resolution $D = 97$, $K = 20$ for the nearest neighbors search and $n = 1000$ averaged images for the CL methods. In Tab. 2, we report the pose correlation (and the average error in radians) after synchronization on three synthetic datasets, with different levels of noise. Tab. 3 reports the correlation and error of the poses after the refinement performed by RELION, as well as the corresponding resolutions (lower is better). We observe that the CL method achieves better initial pose estimation in Tab. 2, although only by a small margin in the low noise regime. However, this does not reflect on the final resolution achieved by RELION in Tab. 3, which is generally lower for CL methods since fewer images are used and averaging loses details. Note also that our MFVDM-$O(2)$ denoising improves the performance of the CL method with respect to MFVDM-$SO(2)$, since it also recovers reflected neighbors and, therefore, allows averaging over a larger set of images with similar viewing direction.

Table 3: Pose correlation, pose error (radians) and estimated resolution after refinement with RELION.

| Synchronization | | VDM (our) | VDM (our) | VDM (our) | CL | CL | CL | CL |
| --- | --- | --- | --- | --- | --- | --- | --- | --- |
| | Denoising | None | $O(2), L=5$ | $O(2), L=10$ | None | $SO(2), L=5$ | $O(2), L=5$ | $O(2), L=10$ |
| N | SNR | | | | | | | |
| 5000 | 0.12 | $99.9_{\pm0.0}$ | $99.8_{\pm0.1}$ | $99.9_{\pm0.0}$ | 76.6 | $85.7_{\pm10.1}$ | $84.2_{\pm9.6}$ | $78.5_{\pm14.2}$ |
| | | $0.04_{\pm0.0}$ | $0.06_{\pm0.02}$ | $0.04_{\pm0.0}$ | — | $0.52_{\pm0.27}$ | $0.50_{\pm0.24}$ | $0.67_{\pm0.35}$ |
| | | $2.42_{\pm0.12}$ | $3.31_{\pm0.63}$ | $2.4_{\pm0.03}$ | 8.0 | $7.65_{\pm3.45}$ | $5.17_{\pm1.47}$ | $9.02_{\pm3.50}$ |
| 5000 | 0.05 | $98.7_{\pm0.0}$ | $99.0_{\pm0.1}$ | $99.0_{\pm0.1}$ | $67.1_{\pm13.2}$ | $59.4_{\pm11.5}$ | $76.4_{\pm2.6}$ | $69.2_{\pm3.3}$ |
| | | $0.10_{\pm0.0}$ | $0.09_{\pm0.01}$ | $0.09_{\pm0.0}$ | $0.93_{\pm0.32}$ | $1.14_{\pm0.32}$ | $0.70_{\pm0.08}$ | $0.91_{\pm0.07}$ |
| | | $3.51_{\pm0.06}$ | $3.47_{\pm0.06}$ | $3.56_{\pm0.0}$ | $6.65_{\pm1.49}$ | $8.21_{\pm4.41}$ | $5.11_{\pm0.83}$ | $6.71_{\pm0.22}$ |

## 6 Conclusions and Discussion

In this work, we studied the symmetries of cryo-EM and found that the $O(2)$ relative poses estimated from the images are sufficient to solve the $SO(3)$ poses synchronization task. With this insight, we proposed a new method which can replace current VDM-based 2D classification methods with limited changes to directly estimate poses rather than clustering and averaging images, while enjoying the same noise robustness properties.

## Acknowledgments and Disclosure of Funding

Funding in direct support of this work: Qualcomm Technologies, Inc.

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
