# APPENDIX

## A   Overview of group representations

In this section we briefly introduce the representation theory of the three groups we used in this work.

**Planar rotations group** $SO(2)$   The standard representation of $r_\theta \in SO(2)$ is as a $2 \times 2$ rotation matrix

$$\rho(r_\theta) = \begin{bmatrix} \cos\theta & -\sin\theta \\ \sin\theta & \cos\theta \end{bmatrix}$$

The complex irreducible representations are often used and correspond to the circular harmonics. More precisely, $SO(2)$ has an irrep for each frequency $k \in \mathbb{Z}$:

$$\sigma_k(r_\theta) = e^{ik\theta} \ .$$

Also, note that $\overline{\sigma_k(r_\theta)} = \sigma_{-k}(r_\theta)$.

**Planar rotations and reflections group** $O(2)$   The standard representation of $O(2)$ is as a $2 \times 2$ orthogonal matrix

$$\rho(r_\theta) = \begin{bmatrix} \cos\theta & -\sin\theta \\ \sin\theta & \cos\theta \end{bmatrix}$$

and

$$\rho(r_\theta f) = \begin{bmatrix} \cos\theta & -\sin\theta \\ \sin\theta & \cos\theta \end{bmatrix} \begin{bmatrix} -1 & 0 \\ 0 & 1 \end{bmatrix}$$

Apart from the *trivial representation* $\rho_{0,0}(h) = 1 \ \forall h \in O(2)$ and the sign-flip representation $\rho_{1,0}(r_\theta) = 1$ and $\rho_{1,0}(f) = -1$, all other irreps are 2 dimensional. For all integer $k > 0$

$$\rho_k(r_\theta) = \begin{bmatrix} \cos k\theta & -\sin k\theta \\ \sin k\theta & \cos k\theta \end{bmatrix}$$

and

$$\rho_k(r_\theta f) = \begin{bmatrix} \cos k\theta & -\sin k\theta \\ \sin k\theta & \cos k\theta \end{bmatrix} \begin{bmatrix} -1 & 0 \\ 0 & 1 \end{bmatrix}$$

**3D rotations group** $SO(3)$   The group $SO(3)$ has irredubile representations $\psi_l$ of frequencies $l = 0, 1, 2, \cdots$ respectively of size $2l + 1$. These representations are isomorphic to the Wigner D matrices. In particular, $\psi_0$ is the trivial representation and $\psi_i$ is isomorphic to the standard representation of $SO(3)$ as $3 \times 3$ rotation matrices.

**3D rotations and reflections group** $O(3)$   The group $O(3)$ is the direct product $SO(3) \times \{e, m_z\}$, where $\{e, m_z\}$ is the group containing the identity and the mirroring in 3D along the $Z$ axis. An element $g = (m, r) \in O(3)$ is a pair of a mirroring $m \in \{e, m_z\}$ and a rotation $r \in SO(3)$. Every irreducible representation $\rho_{i,k}$ of $O(3)$ is equal to the product of an irrep $\rho_k^{SO(3)}$ of $SO(3)$ and one of the two (1 dimensional) irreps $\rho_i^{\{e,m_z\}}$ -for $i = 0, 1$ - of $\{e, m_z\}$:

$$\rho_{i,k}(m, r) = \rho_i^{\{e,m_z\}}(m)\rho_k^{SO(3)}(r)$$

where $\rho_i^{\{e,m_z\}}(e) = 1$ and $\rho_i^{\{e,m_z\}}(m_z) = (-1)^i$.

In general, if $G$ is a group, we denote with $\widehat{G}$ the set of its irreducible representations.

## B   Spectral Properties of the Vector Laplacian on the Projective Plane

Recall the generative process for cryo-EM images:

$$o_i = \Pi(g_i^{-1}\Psi) \quad \text{with } g_i \in SO(3) \tag{12}$$

Let $R_z \cong \mathrm{SO}(2) < \mathrm{SO}(3)$ the subgroup of $\mathrm{SO}(3)$ containing rotations around the $Z$ axis and $H \cong \mathrm{O}(2) < \mathrm{SO}(3)$ the subgroup containing also the rotation $r_y$ by $\pi$ around the $Y$ axis.

Define the projective plane $\mathrm{P}\mathbb{R}^2$ as the quotient space $\mathrm{P}\mathbb{R}^2 \cong \mathrm{SO}(3)/H$. This definition of the projective plane is convenient since the equivalence relationship used corresponds to the symmetries of cryo-EM, as described in Tab. 1. If there exists a $h_{ij} \in H$ such that $o_j \approx h_{ij}.o_i$, the symmetries of the generative model imply that

$$g_j \approx g_i h_{ij}^{-1}$$

and, therefore, that the poses $g_i$ and $g_j$ belong to the same coset in $\mathrm{P}\mathbb{R}^2 \cong \mathrm{SO}(3)/H$. As argued earlier, the estimated relative poses provide a local estimation of the parallel transport on $\mathrm{P}\mathbb{R}^2$.

The parallel transport operator transports vector fields defined over a space. Before entering the details about the parallel transport, let us define more precisely the concept of vector fields.

**Vector fields as Mackey functions**    Let $G' < G$ be two compact groups and $\rho$ a representation of $G'$, i.e. $\rho : G' \to \mathrm{GL}(V_\rho)$, where $V_\rho$ is the vector space where $\rho$ acts (i.e. $\mathbb{R}^{\dim\rho}$ if $\rho$ is a real representation or $\mathbb{C}^{\dim\rho}$ if it is complex valued). Let $X$ be the homogeneous space isomorphic to the quotient $X \cong G/G'$. Then, we define a (square-integrable) $\rho$-vector field over $X$ as a Mackey function, i.e. as a map $v : G \to V_\rho$ satisfying the following $G'$-equivariance property:

$$v(gg') = \rho(g')^{-1}v(g) \quad \forall g \in G, g' \in G' .$$

We denote the space of all $\rho$-vector fields as $\mathrm{Hom}_{G'}(G, V_\rho)$. The space $\mathrm{Hom}_{G'}(G, V_\rho)$ carries a natural action of $G$: let $v$ be a $\rho$-vector field and $g \in G$, then:

$$[g.v](x) = v(g^{-1}.x) \quad \forall x \in G .$$

This representation is isomorphic to the *induced representation* $\mathrm{Ind}_{G'}^G \rho$ and the space of vector fields is isomorphic to $\mathrm{Hom}_{G'}(G, V_\rho) \cong \mathrm{Ind}_{G'}^G V_\rho$; we will use these two notations interchangeably, depending on whether we want to emphasize the $G'$-equivariance property or the representation of $G$ acting on this space. By choosing a *section* of the quotient space, i.e. a function $\gamma : X \to G$ defining a choice of representative element of each coset $gG' \in X$, this Mackey function can be turned into the more familiar notion of vector field $v' : X \to V_\rho$ by composition, i.e. $v' = v \circ \gamma$. A useful property is that, if $G'' < G'$, then $\mathrm{Hom}_{G''}(G, \mathrm{Res}_{G''}^{G'} V_\rho) \subset \mathrm{Hom}_{G'}(G, V_\rho)$. In our particular setting, we are interested in vector fields over $\mathrm{P}\mathbb{R}^2$, on which our local parallel transport operator will act. Hence, $G = \mathrm{SO}(3)$ and $G' = H$. Moreover, we are interested in vector fields of different kinds, i.e. associated with different choices of irreducible representations $\rho$ of $H$.

**The action of $\mathrm{SO}(3)$ on the space of vector fields**    One can show that the action of $G = \mathrm{SO}(3)$, i.e. the representation $\mathrm{Ind}_{\mathrm{O}(2)}^{\mathrm{SO}(3)} \rho$, is unitary and, therefore, *Peter-Weyl Theorem* guarantees that it can be decomposed into a direct sum of (an infinite number of) irreducible representations of $\mathrm{SO}(3)$. In particular, the space of $\rho$-vector fields decomposes into invariant subspaces, each transforming independently under $\mathrm{SO}(3)$ by a different irrep, i.e.:

$$\mathrm{Hom}_H(\mathrm{SO}(3), V_\rho) \cong \bigoplus_{\psi \in \widehat{\mathrm{SO}(3)}} \bigoplus^{m_\psi} V_\psi$$

where $\widehat{\mathrm{SO}(3)}$ is the set of (representatives of the isomorphism classes of) $\mathrm{SO}(3)$'s irreps and $m_\psi$ is the multiplicity of $\psi$ in this decomposition. These invariant subspaces can be identified as follows. By the *Frobenious Reciprocity Theorem*, one can show that the multiplicity of an irrep $\psi$ of $\mathrm{SO}(3)$ into $\mathrm{Ind}_H^{\mathrm{SO}(3)} \rho$ is precisely equal to the multiplicity of $\rho$ in the $\mathrm{Res}_H^{\mathrm{SO}(3)} \psi$, i.e. the representation $\psi$ restricted to the domain $H$. If $\psi$ is a Wigner D matrix of frequency $l$, it is well known that $\mathrm{Res}_H^{\mathrm{SO}(3)} \psi$ decomposes into a direct sum of all 2-dimensional $H \cong \mathrm{O}(2)$ irreps of frequency $k = 1, \cdots, l$ and one of the two 1 dimensional ones, depending on the parity of $l$. It follows that, if $\rho$ has frequency $k > 0$, $\mathrm{Ind}_H^{\mathrm{SO}(3)} \rho$ contains one copy of each irrep of $\mathrm{SO}(3)$ with frequency $l \geq k$.

**A basis for the invariant subspaces of vector fields** Next, we pick a basis for each of the invariant subspaces mentioned above. Together, these bases form a basis for the space of all $\rho$-vector fields. We will state the following results without a proof, but we refer to Appendix D of [32] for a formal discussion. Let $\rho$ be the frequency $k$ irrep of $H \cong O(2)$; then, for any irrep $\psi$ of $SO(3)$ of frequency $l \geq k$, $\text{Ind}_H^{SO(3)} \rho$ contains exactly one subspace $V_\psi$ of dimension $\dim_\psi = 2l + 1$. Let

$$\text{Res}_H^{SO(3)} \psi = [\text{ID}^l]^T \left( \bigoplus_{\rho'} \rho' \right) \text{ID}^l$$

be the irreps decomposition of $\psi$ when restricted to $H$, where the direct sum $\bigoplus$ iterates over the right set of irreps and $\text{ID}^l$ is the appropriate $\dim_\psi \times \dim_\psi$ change of basis. The irrep $\rho$ of frequency $k$ appears in the direct sum above precisely once. Let $\text{ID}^{kl}$ be the $2 \times \dim_\psi$ matrix containing the rows of $\text{ID}^l$ which are acted on by the single occurrence of $\rho$ in the directsum. Then, a basis for $V_\psi$ is given by the following set:

$$\mathcal{B}_l^k = \left\{ v_{li} : SO(3) \to V_\rho, g \mapsto \sqrt{\dim_\psi} \, \text{ID}^{kl} \psi(g^{-1}) e_i \mid i = 1, \ldots, \dim_\psi = 2l + 1 \right\} \quad (13)$$

where $e_i$ is a zero vector containing a 1 in its $i$-th entry. We will now see that these functions are strictly related to the *spin-weighted spherical harmonics*.

**Relation with the spin-weighted spherical harmonics** To see this, we first restrict the 2-dimensional irrep $\rho$ of $O(2)$ with frequency $k > 0$ to the $R_z \cong SO(2)$ subgroup. Let $\sigma$ be the complex irrep of $SO(2)$ of frequency $k$ and $\overline{\sigma}$ the one of frequency $-k$. Then, it holds that $\text{Res}_{SO(2)}^{O(2)} \rho \cong \sigma \oplus \overline{\sigma}$; this isomorphism is given by the matrix

$$C = \frac{1}{\sqrt{2}} \begin{bmatrix} \imath & -1 \\ \imath & 1 \end{bmatrix}, \quad (14)$$

where $\imath$ is the imaginary unit and $\dagger$ represents conjugate transpose. Indeed:

$$\rho(r_\theta r_y^f) = C^\dagger \begin{bmatrix} e^{\imath k\theta} & 0 \\ 0 & e^{-\imath k\theta} \end{bmatrix} \begin{bmatrix} 0 & -1 \\ -1 & 0 \end{bmatrix}^f C = C^\dagger \begin{bmatrix} \sigma(r_\theta) & 0 \\ 0 & \overline{\sigma}(r_\theta) \end{bmatrix} \begin{bmatrix} 0 & -1 \\ -1 & 0 \end{bmatrix}^f C \quad (15)$$

with $f \in \{0, 1\}$ representing respectively no flip or a flip along the $X$ axis and $r_\theta$ a planar rotation by $\theta$. Then, $\text{Hom}_H(SO(3), V_\rho)$ is a subspace of $\text{Hom}_{R_z}(SO(3), \text{Res}_{R_z}^H V_\rho) \cong \text{Hom}_{R_z}(SO(3), V_\sigma) \oplus \text{Hom}_{R_z}(SO(3), V_{\overline{\sigma}})$. The spaces $\text{Ind}_{R_z}^{SO(3)} V_\sigma$ and $\text{Ind}_{R_z}^{SO(3)} V_{\overline{\sigma}}$ are the spaces of $\sigma$ and $\overline{\sigma}$ vector-fields over a sphere $\mathcal{S}^2 \cong SO(3)/R_z$, like those considered in [13]. Each of these spaces contain a single subspace isomorphic to $V_\psi$ for any $\psi$ of frequency $l \geq k$. Let's denote these two spaces as $V_\psi^+$ and $V_\psi^-$; they are, respectively, spanned by the *spin-weighted spherical harmonics* of order $l$ and weights $+k$ and $-k$, i.e.:

$$Y_l^{+k} = \left\{ Y_{li} : SO(3) \to \mathbb{C}, g \mapsto \sqrt{\dim_\psi} e_1^T C \, \text{ID}^{kl} \psi(g^{-1}) e_i \mid i = 1, \ldots, \dim_\psi = 2l + 1 \right\}$$

$$Y_l^{-k} = \left\{ Y_{li} : SO(3) \to \mathbb{C}, g \mapsto \sqrt{\dim_\psi} e_2^T C \, \text{ID}^{kl} \psi(g^{-1}) e_i \mid i = 1, \ldots, \dim_\psi = 2l + 1 \right\}$$

where $e_1 = (1, 0)^T$ and $e_2 = (0, 1)^T$. One can verify that $V_\psi$, the subspace of $\text{Ind}_H^{SO(3)} V_\rho$ spanned by $\mathcal{B}_l^k$, is a $\dim_\psi = 2l + 1$ dimensional subspace of $V_\psi^+ \oplus V_\psi^-$, the $2 \dim_\psi$-dimensional space spanned by the spin weighted spherical harmonics above. This result will turn out useful later when relating the eigenfunctions of the Laplacian operator on the projective plane and on the sphere.

**Eigenfunctions of the Laplacian Operator** By a similar argument used in [13] for the sphere, the *Laplace-Beltrami operator* defined over a $\rho$-vector field can be identified with a local parallel transport operator over the projective space and, therefore, can be shown to be $SO(3)$-equivariant. Because the $SO(3)$ action on the space of vector fields $\text{Hom}_H(SO(3), V_\rho)$ contains at most one copy of each irrep of $SO(3)$, by *Schur's Lemma*, this implies that the operator acts by scalar multiplication in each of the invariant subspaces of the space of $\rho$-vector fields. It follows that these scalar values are the eigenvalues of the operator and the invariant subspaces are the eigenspaces. Additionally, each eigenspace is spanned by the basis $\mathcal{B}_l^k$ defined above for the unique frequency $l \geq k$ such that the eigenspace transforms according to the $SO(3)$'s irrep $\psi$ of frequency $l$. Thus, $\mathcal{B}_l^k$ constitutes a set of *eigen-functions* of the Laplacian operator.

**Localized Parallel Transport over** $\mathrm{P}\mathbb{R}^2$ **and** $\mathcal{S}^2$   Define the sphere as the quotient space $\mathcal{S}^2 \cong \mathrm{SO}(3)/R_z$, with $R_z \cong \mathrm{SO}(2)$. Let $\sigma(r_\theta) = e^{ik\theta}$ be the complex irrep of $\mathrm{SO}(2)$ of frequency $k \in \mathbb{Z}$. Then, let $T_h^{\mathcal{S}^2}$ be the local parallel transport operator over a $\sigma$-vector field $v$ on a sphere $\mathcal{S}^2$ as defined in [13], where $h \in [0, 2]$ defines the locality:

$$[T_h^{\mathcal{S}^2} v](x) := \int_{y \in \mathrm{SO}(3): \langle \pi(y), \pi(x) \rangle > 1-h} \sigma(T^{\mathcal{S}^2}(x, y)) v(y) \quad \forall x \in \mathrm{SO}(3)$$

where $\pi : \mathrm{SO}(3) \to \mathcal{S}^2$ is the projection map defined as $\pi : g = (\boldsymbol{x}, \boldsymbol{y}, \boldsymbol{z}) \to \boldsymbol{z}$. Assume that the spherical cap covered by the locality $h$ is smaller than an hemisphere, i.e. it does not include two points which are projected to the same point in the projective plane. Let $\rho$ be the $H \cong \mathrm{O}(2)$ irrep of frequency $|k|$. Let $T_h^{\mathrm{P}\mathbb{R}^2}$ be the local parallel transport operator of a $\rho$-vector field on $\mathrm{P}\mathbb{R}^2$ defined as

$$[T_h^{\mathrm{P}\mathbb{R}^2} v](x) := \frac{1}{2} \int_{y \in \mathrm{SO}(3): |\langle \pi(y), \pi(x) \rangle| > 1-h} \rho(T^{\mathrm{P}\mathbb{R}^2}(x, y)) v(y)$$

where $T^{\mathrm{P}\mathbb{R}^2}(x, y) = T^{\mathcal{S}^2}(x, y)$ if $\langle \pi(y), \pi(y) \rangle > 0$ and $T^{\mathrm{P}\mathbb{R}^2}(x, y) = T^{\mathcal{S}^2}(x, yr_y)r_y$ otherwise ($r_y \in H \cong \mathrm{O}(2)$ is the $\pi$ rotation around the $Y$ axis). Note that $\pi(yr_y) = -\pi(y)$. Then, it follows that:

$$[T_h^{\mathrm{P}\mathbb{R}^2} v](x) = \frac{1}{2} \int_{y \in \mathrm{SO}(3): \langle \pi(y), \pi(x) \rangle > 1-h} \rho(T^{\mathcal{S}^2}(x, y)) v(y)$$
$$+ \frac{1}{2} \int_{y \in \mathrm{SO}(3): \langle -\pi(y), \pi(x) \rangle > 1-h} \rho(T^{\mathcal{S}^2}(x, yr_y)) \rho(r_y) v(y)$$

applying the change of variables $y' = yr_y$ and using $r_y r_y = e$:

$$= \frac{1}{2} \int_{y \in \mathrm{SO}(3): \langle \pi(y), \pi(x) \rangle > 1-h} \rho(T^{\mathcal{S}^2}(x, y)) v(y)$$
$$+ \frac{1}{2} \int_{y' \in \mathrm{SO}(3): \langle \pi(y'), \pi(x) \rangle > 1-h} \rho(T^{\mathcal{S}^2}(x, y')) \rho(r_y) v(y' r_y)$$
$$= [T_h^{\mathcal{S}^2} \frac{1}{2} [v + r_y \triangleleft v]](x)$$

where $r_y \triangleleft v$ is the $\rho$-vector field defined as $[r_y \triangleleft v](x) = \rho(r_y) v(x r_y)$. If $v : \mathrm{SO}(3) \to \mathbb{R}^{\dim_\rho}$ is a $\rho$-vector field over $\mathrm{P}\mathbb{R}^2$, the equivariance property of Mackey functions implies that $r_y \triangleleft v = v$. It follows that

$$T_h^{\mathrm{P}\mathbb{R}^2} v = T_h^{\mathcal{S}^2} v^{\uparrow}$$

where $v^{\uparrow}$ is the lifting of $v$ over the sphere $\mathcal{S}^2$ in order to use $T_h^{\mathcal{S}^2}$.

In other words, *the parallel transport operator $T_h^{\mathrm{P}\mathbb{R}^2}$ over $\mathrm{P}\mathbb{R}^2$ is equivalent to the parallel transport operator $T_h^{\mathcal{S}^2}$ over $\mathcal{S}^2$ with its domain restricted to $r_y$-invariant vector fields on the sphere* . This enables us to derive the eigenvalues of $T_h^{\mathrm{P}\mathbb{R}^2}$ from the eigenvalues of $T_h^{\mathcal{S}^2}$.

**Eigenvalues of the Localized Parallel Transport operator over** $\mathrm{P}\mathbb{R}^2$   We have already argued that the local parallel transport operator $T_h^{\mathrm{P}\mathbb{R}^2}$ over $\rho$-vector fields, with $\rho$ the frequency $k$ irreps of $H \cong \mathrm{O}(2)$, has one eigenspace $V_\psi$ transforming according to the irrep $\psi \in \widehat{\mathrm{SO}(3)}$ of frequency $l$ for all $l \geq k$. We also know that $V_\psi$ is a subspace of $V_\psi^+ \oplus V_\psi^-$, the corresponding eigenspaces of the operator $T_h^{\mathcal{S}^2}$ defined respectively over $\sigma$ and $\overline{\sigma}$ vector fields, with $\sigma$ the frequency $k$ irrep of $R_z \cong \mathrm{SO}(2)$. The lifting operator $\cdot^{\uparrow} : \mathrm{Hom}_H(G, V_\rho) \to \mathrm{Hom}_{R_z}(G, \mathrm{Res}_{R_z}^H V_\rho)$ used earlier embeds each $V_\psi$ into $V_\psi^+ \oplus V_\psi^-$. Finally, note that the operator $T_h^{\mathcal{S}^2}$ acts on $V_\psi^+$ and $V_\psi^-$ independently by multiplying each of them by their own eigenvalue. However, the eigenvalues are real numbers and, since $T_h^{\mathcal{S}^2}$ over $\sigma$ vector fields and $T_h^{\mathcal{S}^2}$ over $\overline{\sigma}$ vector-fields are equal up to conjugation, both operators share the same eigenvalues. It follows that $T_h^{\mathrm{P}\mathbb{R}^2}$ acts by multiplying a subspace $V_\psi$ by the eigenvalue of $T_h^{\mathcal{S}^2}$ associated with $V_\psi^+$ (or equivalently $V_\psi^-$). Thus, *the operators $T_h^{\mathrm{P}\mathbb{R}^2}$ and $T_h^{\mathcal{S}^2}$ share the same eigenvalues.*

The eigenvalues $\{\lambda_l^k\}_{l \geq |k|}$ of the operator $T_h^{\mathcal{S}^2}$ defined over $\sigma$ (or $\overline{\sigma}$) vector-fields were already derived in [13]. In particular, this implies that the same analysis of the spectral gap performed in [13] applies here.

In conclusion, we note that the difference between using the operator $T_h^{\mathbb{PR}^2}$ on $\mathrm{Hom}_H(\mathrm{SO}(3), V_\rho)$ and $T_h^{\mathcal{S}^2}$ on $\mathrm{Hom}_{R_z}(\mathrm{SO}(3), V_\sigma \oplus V_{\overline{\sigma}})$ is only in the dimensionality of the eigenspaces, which are twice larger in the second case. It is inside this additional degree of freedom that the *global* $\mathrm{SO}(2)$ *ambiguity* we introduced in Section 2 lies.

## C  SO(3) **pose synchronization with** O(2) **relative poses**

In Section 2 we claimed that $H \cong \mathrm{O}(2)$ relative poses provide sufficient constraints to solve the $\mathrm{SO}(3)$ synchronization problem. This is the same idea behind the final synchronization step in Section 3.2. In this section, we formalize and prove this claim.

**The simplified synchronization problem**   First, note that an element $g = (\boldsymbol{x}, \boldsymbol{y}, \boldsymbol{z}) \in \mathrm{SO}(3)$ is fully identified by the matrix $(\boldsymbol{x}, \boldsymbol{y}) \in \mathbb{R}^{3 \times 2}$ since $\boldsymbol{z} = \boldsymbol{x} \times \boldsymbol{y}$. Now, assume that all relative poses are elements of $H \cong \mathrm{O}(2)$, i.e. $\{\widehat{g}_{ij} = h_{ij} \in H\}_{ij}$. Note that any element $h \in H$ has form

$$h = \begin{bmatrix} \rho(h) & \boldsymbol{0} \\ \boldsymbol{0}^T & \det(\rho(h)) \end{bmatrix}$$

where $\rho(h)$ is the representation of $h$ as the standard $2 \times 2$ orthogonal matrix, $\boldsymbol{0}$ is a 2-dimensional vector containing zeros and $\det(\rho(h)) = \pm 1$.

Then, $g_j = g_i h_{ij} \iff (\boldsymbol{x}_j, \boldsymbol{y}_j) = (\boldsymbol{x}_i, \boldsymbol{y}_i)\rho(h_{ij})$. The proof of $\implies$ is trivial. To prove $\impliedby$, note that

$$\begin{aligned} \langle \boldsymbol{z}_j, \boldsymbol{z}_i \rangle &= \langle \boldsymbol{x}_j \times \boldsymbol{y}_j, \boldsymbol{x}_i \times \boldsymbol{y}_i \rangle \\ &= \det\left((\boldsymbol{x}_j, \boldsymbol{y}_j)^T (\boldsymbol{x}_i, \boldsymbol{y}_i)\right) \\ &= \det\left(\rho(h_{ij})^T (\boldsymbol{x}_i, \boldsymbol{y}_i)^T (\boldsymbol{x}_i, \boldsymbol{y}_i)\right) \\ &= \det\left(\rho(h_{ij})\right) \det\left((\boldsymbol{x}_i, \boldsymbol{y}_i)^T (\boldsymbol{x}_i, \boldsymbol{y}_i)\right) \\ &= \det\left(\rho(h_{ij})\right) \end{aligned}$$

Since both $\boldsymbol{z}_i$ and $\boldsymbol{z}_j$ are unit vectors, it follows that $\boldsymbol{z}_j = \det\left(\rho(h_{ij})\right) \boldsymbol{z}_i$ and, therefore, $g_j = g_i h_{ij}$.

This result suggests that, when the relative poses belong to $H$ (or a subgroup of $H$), the synchronization problem in Eq. 2 can be simplified from the set of constraints

$$\widehat{g}_j \approx \widehat{g}_i \widehat{h}_{ij}^{-1} \quad \forall(i,j) \in \mathcal{G} \tag{16}$$

to the following ones

$$(\widehat{\boldsymbol{x}}_j, \widehat{\boldsymbol{y}}_j) \approx (\widehat{\boldsymbol{x}}_i, \widehat{\boldsymbol{y}}_i)\rho(\widehat{h}_{ij}^{-1}) \quad \forall(i,j) \in \mathcal{G} \ . \tag{17}$$

**Spectral Relaxation of the simplified synchronization problem**   Finally, we show that the simplified synchronization problem in Eq. 17 is related to the eigenvalue decomposition of the graph Connection Laplacian associated with the frequency $k = 1$ representation of $\mathrm{O}(2)$ via a *spectral relaxation*.

First, we formally define the synchronization problem as

$$\begin{aligned} \{(\widehat{\boldsymbol{x}}_i, \widehat{\boldsymbol{y}}_i)\}_i &= \underset{\{(\boldsymbol{x}_i, \boldsymbol{y}_i) | \boldsymbol{x}_i \perp \boldsymbol{y}_i \in \mathcal{S}^2\}_i}{\arg\max} \mathcal{L}(\{(\boldsymbol{x}_i, \boldsymbol{y}_i)\}_i) \\ &= \underset{\{(\boldsymbol{x}_i, \boldsymbol{y}_i) | \boldsymbol{x}_i \perp \boldsymbol{y}_i \in \mathcal{S}^2\}_i}{\arg\max} \sum_{(i,j) \in \mathcal{G}} \frac{1}{\sqrt{\deg(i)\deg(j)}} \mathrm{Tr}\left((\boldsymbol{x}_j, \boldsymbol{y}_j)^T (\boldsymbol{x}_i, \boldsymbol{y}_i)\rho(h_{ij})^T\right) \end{aligned}$$

where we require $\boldsymbol{x}_i$ and $\boldsymbol{y}_i$ to be orthogonal unit vectors in order to represent an element of $\mathrm{SO}(3)$. $\deg(i)$ is the degree of the node $i$ in the graph $\mathcal{G}$. One can show that maximizing this objective function is equivalent to minimizing the squared Frobenious distance between $(\boldsymbol{x}_j, \boldsymbol{y}_j)$ and $(\boldsymbol{x}_i, \boldsymbol{y}_i)\rho(h_{ij})^T$.

The objective $\mathcal{L}(\{(\boldsymbol{x}_i, \boldsymbol{y}_i)\}_i)$ can be re-written in matrix form. Define the matrix $V \in \mathbb{R}^{2N \times 3}$ as the matrix whose $i$-th $2 \times 3$ block contains $(\boldsymbol{x}_i, \boldsymbol{y}_i)^T$ and let $V_x, V_y, V_z$ be its 3 columns. Let $A$ be a $2N \times 2N$ symmetric matrix representing the normalized graph Connection Laplacian associated with the representation $\rho$ of $H$ and the graph $\mathcal{G}$. Then, the objective above can be expressed as

$$\mathcal{L}(\{(\boldsymbol{x}_i, \boldsymbol{y}_i)\}_i) = \mathrm{Tr}\left(V^T A V\right) = V_x^T A V_x + V_y^T A V_y + V_z^T A V_z \ .$$

One recognizes the similarity between maximizing $\mathcal{L}(\{(\boldsymbol{x}_i, \boldsymbol{y}_i)\}_i)$ and the eigenvalues problem. Indeed, if we drop the condition that $\boldsymbol{x}_i$ and $\boldsymbol{y}_i$ need to be orthogonal unit vectors, this objective is maximized by any choice of $V_x, V_y, V_z$ in the top eigenspace of $A$. This is the idea behind the *spectral relaxation* of the problem.

**Convergence of the spectral relaxation to the true poses** In practice, we know from Appendix B that the top eigenspace of $A$ is precisely 3 dimensional. Moreover, we know from Eq. 13 that this eigenspace is spanned (up to a normalizing factor) by the three eigenvectors

$$\mathcal{B}_1^1 = \left\{ v_{1,a} : \mathcal{V} \to \mathbb{R}^2, i \mapsto \sqrt{3}\, \mathrm{ID}^{11}\, \psi_1(g_i^{-1}) e_a \mid a = 1, 2, 3 \right\} \tag{18}$$

where $\psi_1 : \mathrm{SO}(3) \to \mathbb{R}^{3 \times 3}$ is the frequency 1 irrep of $\mathrm{SO}(3)$ which is isomorphic to the Wigner D Matrix of frequency 1 as well as the standard representation of $\mathrm{SO}(3)$ are $3 \times 3$ rotation matrices. Recall that the matrix $\mathrm{ID}^{11}$ contains the rows of the change of basis $\mathrm{ID}^1$ in the decomposition $\mathrm{Res}_H^{\mathrm{SO}(3)} \psi_1(h) = (\mathrm{ID}^1)^T (\rho(h) \oplus \det(\rho(h)))\, \mathrm{ID}^1$ . Moreover, $\rho(h) \oplus \det(\rho(h))$ is also the representation of $H$ acting on the standard representation of the elements of $\mathrm{SO}(3)$ as a $3 \times 3$ rotation matrix $(\boldsymbol{x}, \boldsymbol{y}, \boldsymbol{z})$. Hence, $\mathrm{ID}^1 \psi_1(g_i)(\mathrm{ID}^1)^T = (\boldsymbol{x}_i, \boldsymbol{y}_i, \boldsymbol{z}_i)$.

However, the eigenvalues decomposition can converge to any orthogonal basis for this space. More precisely, let $v : \mathcal{V} \to \mathbb{R}^{2 \times 3}$ the stack of the three elements in $\mathcal{B}_1^1$ and let $\varphi_1 : \mathcal{V} \to \mathbb{R}^{2 \times 3}$ the top three eigenvectors found. Then, there is an, unknown, matrix $O \in \mathrm{O}(3)$ such that $\varphi_1(i) = v(i) O$ and, therefore, $\varphi_1(i) = \sqrt{3}(\boldsymbol{x}_i, \boldsymbol{y}_1)^T O$. This guarantees the recovery of the frames, and so, of the poses $\{g_i\}_i$ up to a global $\mathrm{O}(3)$ ambiguity whenever the graph Connection Laplacian converges to the Laplacian operator over $\mathrm{P}\mathbb{R}^2$.

This global $\mathrm{O}(3)$ ambiguity is related to the intrinsic $\mathrm{O}(3)$ ambiguity in cryo-EM. Indeed, let $O = r\, \mathrm{i}^c \in \mathrm{O}(3)$, where $r \in \mathrm{SO}(3)$ and $c \in \{0, 1\}$ indicates whether $O$ includes the inversion $\mathrm{i} = -1I$. Note that $\mathrm{i}\,\boldsymbol{x} = -\boldsymbol{x}$ and $\det(O) = 1 - 2c$ and $\det(O)O = r$. By using

$$Ox(i) \times Oy(i) = \det(O)\, O\left(x(i) \times y(i)\right) = r\left(x(i) \times y(i)\right) \ ,$$

one can verify that $O$ maps a solution

$$\{\widehat{g}_i = (x(i), y(i), z(i))\}_i$$

to

$$\{r\widehat{g}_i r_z^c = (\mathrm{i}^c\, rx(i), \mathrm{i}^c\, ry(i), rz(i))\}_i \ .$$

## C.1 Failure with $\mathrm{SO}(2)$ relative poses

Finally, we emphasize the necessity of using $\mathrm{O}(2)$ rather than $\mathrm{SO}(2)$ relative poses. Let $\rho \in \widehat{\mathrm{O}(2)}$ be again the 2-dimensional real irrep of $\mathrm{O}(2)$ of frequency $k > 0$ and recall that $\mathrm{Res}_{\mathrm{O}(2)}^{\mathrm{SO}(2)} \rho \cong \sigma \oplus \overline{\sigma}$, where $\sigma : r_\theta \to e^{ik\theta}$ is the frequency $k$ complex irrep of $\mathrm{SO}(2)$; see Eq. 15. As argued in Apx. B, for $\psi \in \widehat{\mathrm{SO}(3)}$, an eigenspace $V_\psi$ of a parallel transport operator $T_h^{\mathrm{P}\mathbb{R}^2}$ over $\rho$-vector fields on the projective plane is a subspace of $V_\psi^+ \oplus V_\psi^-$, which are the corresponding eigenspaces of the operator $T_h^{\mathcal{S}^2}$ over $\sigma$ and $\overline{\sigma}$ vector fields over the sphere. In case the synchronization graph only contains $\mathrm{SO}(2)$ relative poses, the operator $T_h^{\mathrm{P}\mathbb{R}^2}$ over $\rho$-vector fields on the projective plane decomposes into two independent copies of $T_h^{\mathcal{S}^2}$ acting over the two subspaces containing $\sigma$ and $\overline{\sigma}$ vector fields on the sphere. In Apx. B, we have also proved that $V_\psi^+$ and $V_\psi^-$ share the same eigenvalues. It follows that these eigenspaces can not be discriminated by the eigenvalues decomposition and the top eigenspace of the synchronization matrix $A$ will be 6 rather than 3 dimensional. This prevents the identification of the absolute poses, which we have proved to live in the 3-dimensional subspace $V_\psi$.

# D   Proof of Theorem 3.1

Assume the eigenvectors of graph connection Laplacian matrices converged to the eigenfunctions of the vector diffusion operators on the projective plane. Let $\rho \in \widehat{O(2)}$ be the 2-dimensional real irrep of $O(2)$ of frequency $k$. Then, the top eigen-space of $\widehat{A_k}$ is $2k+1$ dimensional with eigenvectors equal to the eigenfunctions defined in Eq. 13 up to an orthogonal change of basis $M \in \mathbb{R}^{2k+1 \times 2k+1}$. In other words, let $Y_k : SO(3) \to \mathbb{R}^{2 \times 2k+1}$ the stack of all $2k+1$ eigenfunctions in Eq. 13; then the top $2k+1$ eigenvectors will have form:

$$\varphi_i^k = \varphi^k(g_i) = Y_k(g_i)M = \sqrt{2k+1}\,\mathrm{ID}^{kk}\,\psi(g_i^{-1})M \in \mathbb{R}^{2 \times 2k+1} \tag{19}$$

for each image $o_i$ with pose $g_i \in SO(3)$.

Then, the $2 \times 2$ block $\widehat{A_k}(i,j)$ is equal to

$$
\begin{aligned}
\widehat{A_k}(i,j) &= \varphi_j^k (\varphi_i^k)^T \\
&= (2k+1)\,\mathrm{ID}^{kk}\,\psi(g_j^{-1})MM^T\psi(g_i)(\mathrm{ID}^{kk})^T \\
&= (2k+1)\,\mathrm{ID}^{kk}\,\psi(g_j^{-1}g_i)(\mathrm{ID}^{kk})^T \\
&= (2k+1)\,\mathrm{ID}^{kk}\,\psi(g_{ij})(\mathrm{ID}^{kk})^T
\end{aligned}
$$

It is now convenient to express the irrep $\psi$ in a *complex* basis. Let $g = (\alpha, \theta, \gamma) \in SO(3)$ expressed in terms of Euler angles and let $d_{mn}^l : [0, \pi] \to \mathbb{C}$ be Wigner's small d function. Then, if $\psi$ is the frequency $l$ irrep of $SO(3)$, $\psi$ is isomorphic to the Wigner D matrix $D^l$ defined as:

$$D_{mn}^l(\alpha, \theta, \gamma) = e^{-im\alpha}d_{mn}^l(\theta)e^{-in\gamma}$$

where $m, n = -l, \ldots, -1, 0, 1, \ldots, l$ index the entries of $D^l$. Let $B$ be the change of basis such that $\psi = BD^lB^\dagger$ and let $C$ be the change of basis matrix defined in Eq. 14.

Since both the Frobenious norm and the determinant are invariant to unitary change of basis, we can consider the block $\underline{\widehat{A_k}}(i,j) = C^\dagger \widehat{A_k}(i,j)C$ instead. Using the changes of basis above:

$$
\begin{aligned}
\underline{\widehat{A_k}}(i,j) &= C^\dagger \widehat{A_k}(i,j)C \\
&= (2k+1)C^\dagger\,\mathrm{ID}^{kk}\,BD^k(g_{ij})B^\dagger(\mathrm{ID}^{kk})^TC
\end{aligned}
$$

Note that the matrix $B^\dagger(\mathrm{ID}^{kk})^TC$ contains the rows of the irreps decomposition of $\mathrm{Res}_H^{SO(3)}\,D^k$ corresponding to the irrep $C\rho C^\dagger$. Recall that

$$C\rho(r_\gamma r_y^f)C^\dagger = \begin{bmatrix} e^{ik\gamma} & 0 \\ 0 & e^{-ik\gamma} \end{bmatrix} \begin{bmatrix} 0 & -1 \\ -1 & 0 \end{bmatrix}^f.$$

Hence, $C\rho C^\dagger$ acts directly on the columns $n = \pm k$ of $D^l$. This implies that $B^\dagger(\mathrm{ID}^{kk})^TC$ simply indexes these two columns; then

$$
\begin{aligned}
\underline{\widehat{A_k}}(i,j) &= (2k+1)C^\dagger\,\mathrm{ID}^{kk}\,BD^l(g_{ij})B^\dagger(\mathrm{ID}^{kk})^TC \\
&= (2k+1)\begin{bmatrix} D_{-k-k}^k(g_{ij}) & D_{-kk}^k(g_{ij}) \\ D_{k-k}^k(g_{ij}) & D_{kk}^k(g_{ij}) \end{bmatrix}
\end{aligned}
$$

Let $g_{ij} = (\alpha, \theta, \gamma)$, where $\cos\theta = \langle z_i, z_j \rangle$ (see Eq. 55 in [13]). Then:

$$
\begin{aligned}
\underline{\widehat{A_k}}(i,j) &= (2k+1)\begin{bmatrix} e^{-ik\alpha} & 0 \\ 0 & e^{ik\alpha} \end{bmatrix}\begin{bmatrix} d_{-k-k}^k(\theta) & d_{-kk}^k(\theta) \\ d_{k-k}^k(\theta) & d_{kk}^k(\theta) \end{bmatrix}\begin{bmatrix} e^{-ik\gamma} & 0 \\ 0 & e^{ik\gamma} \end{bmatrix} \\
&= (2k+1)\begin{bmatrix} e^{-kl(\alpha+\gamma)}d_{-k-k}^k(\theta) & e^{-ik(\alpha-\gamma)}d_{-kk}^k(\theta) \\ e^{ik(\alpha-\gamma)}d_{k-k}^k(\theta) & e^{ik(\alpha+\gamma)}d_{kk}^k(\theta) \end{bmatrix}
\end{aligned}
$$

$$\left\|\underline{\widehat{A_k}}(i,j)\right\|_F^2 = \left\|d_{-k-k}^k(\theta)\right\|^2 + \left\|d_{-kk}^k(\theta)\right\|^2 + \left\|d_{k-k}^k(\theta)\right\|^2 + \left\|d_{kk}^k(\theta)\right\|^2$$

$$\det\left(\underline{\widehat{A_k}}(i,j)\right) = d_{-k-k}^k(\theta)d_{kk}^k(\theta) - d_{-kk}^k(\theta)d_{k-k}^k(\theta)$$

By expanding the definition of the Wigner small d functions $d^k$, one can show that:

$$d^k_{kk}(\theta) = d^k_{k-k}(\theta) = (\cos\frac{\theta}{2})^{2k}$$

$$d^k_{-kk}(\theta) = d^k_{k-k}(\theta) = (\sin\frac{\theta}{2})^{2k}$$

and, therefore:

$$\left\|\widehat{A_k}(i,j)\right\|_F^2 = 2(\cos\frac{\theta}{2})^{4k} + 2(\sin\frac{\theta}{2})^{4k}$$

$$\det\left(\widehat{A_k}(i,j)\right) = (\cos\frac{\theta}{2})^{4k} - (\sin\frac{\theta}{2})^{4k}$$

Finally, by using the trigonometric identities $\sin^2\frac{\alpha}{2} = \frac{1-\cos\alpha}{2}$ and $\cos^2\frac{\alpha}{2} = \frac{1+\cos\alpha}{2}$, one reaches the identities in Theorem 3.1.

# E  Similarity as dot product for fast $K$-NN search

In Sec. 3, the way the estimator $\widehat{s}^\pm_{ij}$ aggregates the multi-frequency information was inspired from [13]. Instead, here, we use an estimator for the similarities similar to the one adopted in [12]:

$$\bar{s}^\pm_{ij} = \sum_{k=1}^{L} S^{k\pm}_{ij}$$

$$\bar{s}_{ij} = \text{sign}(\bar{s}^+_{ij} - \bar{s}^-_{ij}) \cdot \max(\bar{s}^+_{ij}, \bar{s}^-_{ij})$$

i.e., we just sum the estimators associated with each frequency. The benefit of this solution is that this estimation can be written as an inner product, enabling faster $K$-NN search.

Indeed, the quantity $S^{k\pm}_{ij}$ itself can be expressed as the inner product of some features $\phi_i$ and $\phi_j$ associated with the images $i$ and $j$. This enables the use of a faster method to identify the $K$ nearest neighbors of each image $i$, without computing the similarity of all pairs.

Recall that we defined $\underline{\varphi_k}(i) = \frac{\sqrt{2}}{\|\varphi_k(i)\|_F}\varphi_k(i) \in \mathbb{R}^{2k+1\times2}$ and $\widehat{A_k}(i,j) = \underline{\varphi_k}(i)^T\underline{\varphi_k}(j) \in \mathbb{R}^{2\times2}$. Theorem 3.1 and Eq. 8 require the Frobenius norm $\left\|\widehat{A_k}(i,j)\right\|_F^2$ and the determinant $\det\left(\widehat{A_k}(i,j)\right)$ of this matrix. As discussed in Sec. 3, in practice, more eigenvectors can be used; in this case $\underline{\varphi_k}(i) \in \mathbb{R}^{d_k\times2}$, with $d_k > 2k+1$.

To simplify our notation, we denote with $\mathrm{x}_{ki}$ and $\mathrm{y}_{ki} \in \mathbb{R}^{d_k}$ the two columns of $\underline{\varphi_k}(i)$, such that $\widehat{A_k}(i,j) = \begin{bmatrix}\mathrm{x}_{ki}^T \\ \mathrm{y}_{ki}^T\end{bmatrix}\begin{bmatrix}\mathrm{x}_{kj} & \mathrm{y}_{kj}\end{bmatrix}$. Moreover, $\langle A, B\rangle = \text{Tr}\left(A^T B\right)$ denotes the standard inner product between matrices and $A \oplus B$ stacks the two matrices. Then, the following identities hold:

$$\left\|\widehat{A_k}(i,j)\right\|_F^2 = (\mathrm{x}_{ki}^T\mathrm{x}_{kj})^2 + (\mathrm{x}_{ki}^T\mathrm{y}_{kj})^2 + (\mathrm{y}_{ki}^T\mathrm{x}_{kj})^2 + (\mathrm{y}_{ki}^T\mathrm{y}_{kj})^2$$

$$= \text{Tr}\left(\mathrm{x}_{ki}\mathrm{x}_{ki}^T\mathrm{x}_{kj}\mathrm{x}_{kj}^T\right) + \text{Tr}\left(\mathrm{x}_{ki}\mathrm{x}_{ki}^T\mathrm{y}_{kj}\mathrm{y}_{kj}^T\right) + \text{Tr}\left(\mathrm{y}_{ki}\mathrm{y}_{ki}^T\mathrm{x}_{kj}\mathrm{x}_{kj}^T\right) + \text{Tr}\left(\mathrm{y}_{ki}\mathrm{y}_{ki}^T\mathrm{y}_{kj}\mathrm{y}_{kj}^T\right)$$

$$= \left\langle\mathrm{x}_{ki}\mathrm{x}_{ki}^T + \mathrm{y}_{ki}\mathrm{y}_{ki}^T, \mathrm{x}_{kj}\mathrm{x}_{kj}^T + \mathrm{y}_{kj}\mathrm{y}_{kj}^T\right\rangle$$

$$\det\left(\widehat{A_k}(i,j)\right) = (\mathrm{x}_{ki}^T\mathrm{x}_{kj})(\mathrm{y}_{ki}^T\mathrm{y}_{kj}) - (\mathrm{x}_{ki}^T\mathrm{y}_{kj})(\mathrm{y}_{ki}^T\mathrm{x}_{kj})$$

$$= \text{Tr}\left(\mathrm{y}_{ki}\mathrm{x}_{ki}^T\mathrm{x}_{kj}\mathrm{y}_{kj}^T\right) - \text{Tr}\left(\mathrm{y}_{ki}\mathrm{x}_{ki}^T\mathrm{y}_{kj}\mathrm{x}_{kj}^T\right)$$

$$= \left\langle\mathrm{x}_{ki}\mathrm{y}_{ki}^T, \mathrm{x}_{kj}\mathrm{y}_{kj}^T - \mathrm{y}_{kj}\mathrm{x}_{kj}^T\right\rangle$$

$$S^{k\pm}_{ij} = \frac{1}{4}\left\|\widehat{A_k}(i,j)\right\|_F^2 \pm \frac{1}{2}\det\left(\widehat{A_k}(i,j)\right)$$

$$= \frac{1}{4}\left\langle\mathrm{x}_{ki}\mathrm{x}_{ki}^T + \mathrm{y}_{ki}\mathrm{y}_{ki}^T, \mathrm{x}_{kj}\mathrm{x}_{kj}^T + \mathrm{y}_{kj}\mathrm{y}_{kj}^T\right\rangle \pm \frac{1}{2}\left\langle\mathrm{x}_{ki}\mathrm{y}_{ki}^T, \mathrm{x}_{kj}\mathrm{y}_{kj}^T - \mathrm{y}_{kj}\mathrm{x}_{kj}^T\right\rangle$$

$$= \frac{1}{2}\left\langle\frac{1}{\sqrt{2}}\left(\mathrm{x}_{ki}\mathrm{x}_{ki}^T + \mathrm{y}_{ki}\mathrm{y}_{ki}^T\right) \oplus \left(\mathrm{x}_{ki}\mathrm{y}_{ki}^T\right), \frac{1}{\sqrt{2}}\left(\mathrm{x}_{kj}\mathrm{x}_{kj}^T + \mathrm{y}_{kj}\mathrm{y}_{kj}^T\right) \oplus \left(\pm\mathrm{x}_{kj}\mathrm{y}_{kj}^T \mp \mathrm{y}_{kj}\mathrm{x}_{kj}^T\right)\right\rangle$$

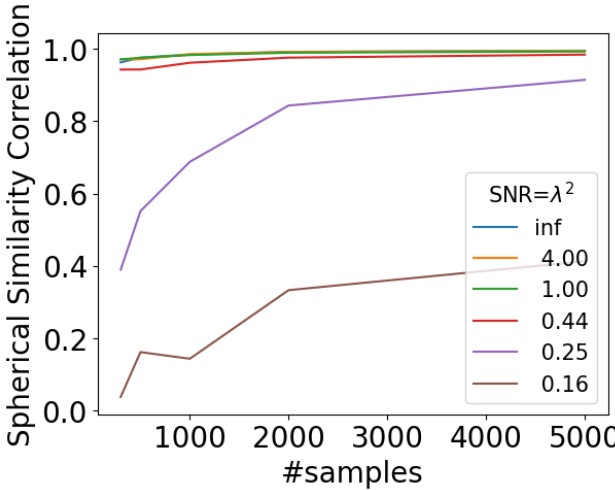

Figure 6: Correlation of the estimated cosine similarities $s_{ij}$ in different variations of the synthetic vector dataset.

Hence, for frequency $k$, we define a *query* vector as $\boldsymbol{q}_{ki} = \frac{1}{\sqrt{2}} \left( \mathrm{x}_{ki}\mathrm{x}_{ki}^T + \mathrm{y}_{ki}\mathrm{y}_{ki}^T \right) \oplus \left( \mathrm{x}_{ki}\mathrm{y}_{ki}^T \right)$ and a *key* vector as $\boldsymbol{k}_{ki}^{\pm} = \frac{1}{\sqrt{2}} \left( \mathrm{x}_{ki}\mathrm{x}_{ki}^T + \mathrm{y}_{ki}\mathrm{y}_{ki}^T \right) \oplus \left( \pm \mathrm{x}_{ki}\mathrm{y}_{ki}^T \mp \mathrm{y}_{ki}\mathrm{x}_{ki}^T \right)$. The final *query* and *key* vectors are obtained by stacking the ones built for each frequency, i.e. $\boldsymbol{q}_i = \bigoplus_k^L \boldsymbol{q}_{ki}$ and $\boldsymbol{k}_i^{\pm} = \bigoplus_k^L \boldsymbol{v}_{ki}^{\pm}$. The indexes $j$ of the key vectors $\boldsymbol{k}_j^{\pm}$ closest to $\boldsymbol{q}_i$ correspond to the indexes with highest similarity $\bar{s}_{ij}^{\pm}$.

Our final estimator is the maximum of the two similarities above (indicated by $\pm$), i.e. $\bar{w}_{ij} = |\bar{s}_{ij}| = \max(\bar{s}_{ij}^+, \bar{s}_{ij}^-)$, where the sign $\pm$ indicates whether the images $i$ and $j$ are related by a reflection. Hence, once can just run $K$-NN on the augmented keys set, containing both $\boldsymbol{k}_j^+$ and $\boldsymbol{k}_j^-$.

Finally, the complexity of $K$-NN is $O(NKd \log N)$, where $d$ is the dimensionality of queries and keys and $N$ the number of points. In this case, assuming at most $M$ eigenvectors are used for any frequency (i.e. for each $k$, $d_k < M$), the dimensionality $d$ is at most $2LM^2$. Hence, the cost of this search is $O(NKLM^2 \log N)$.

# F  Additional Experiments and Details

## F.1  Synthetic Vector Dataset

In this section, we provide further details and results on the datasets used in Sec. 5.1.

Figure 6 shows the correlation between the estimated cosine similarities $\{\tilde{s}_{ij}\}_{ij}$ and the real ones $\{s_{ij}\}_{ij}$ for the same set of experiments shown in Figure 4. The same observations hold here. In these experiments, we used $L = 6$.

In Figure 7, we compare the correlation between the estimated and ground truth $SO(3)$ poses in different experimental settings at a $SNR = 0.16$. While the noise is too large to achieve perfect correlation with the number of samples considered, we see that there is a regime where considering more frequencies significantly improves the performance. A similar effect is observed on the estimation of the cosine similarity in Figure 8. Note that in these experiments, as well as in Sec. 5.1 and Figure 5, we used the estimator $\tilde{s}_{ij}$ from Eq. 9 rather than $\hat{s}_{ij}$ from Eq. 10 since $\tilde{s}_{ij}$ directly estimates $s_{ij}$, while $\hat{s}_{ij}$ is equal to either $\hat{s}_{ij}^+$ or $-\hat{s}_{ij}^-$ and matches $s_{ij}$ well only when $|s_{ij}|$ is close to 1.

## F.2  Further details on the experiments with ASPIRE and RELION in Sec. 5.2

Here, we provide more details about the cryo-EM experiments in Sec. 5.2.

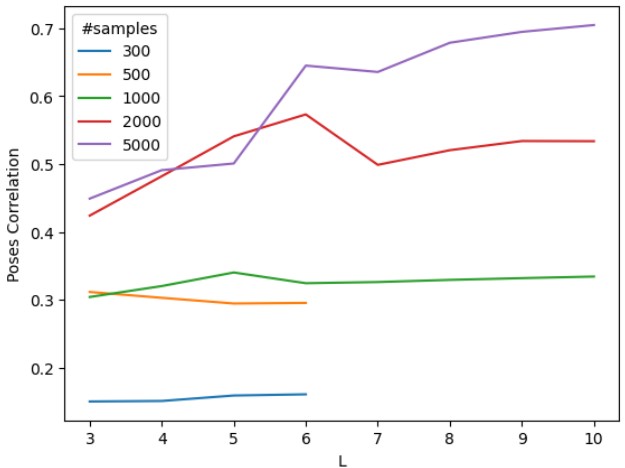

Figure 7: Effect of different choices of maximum frequency $L$ on the performance with $SNR = 0.16$.

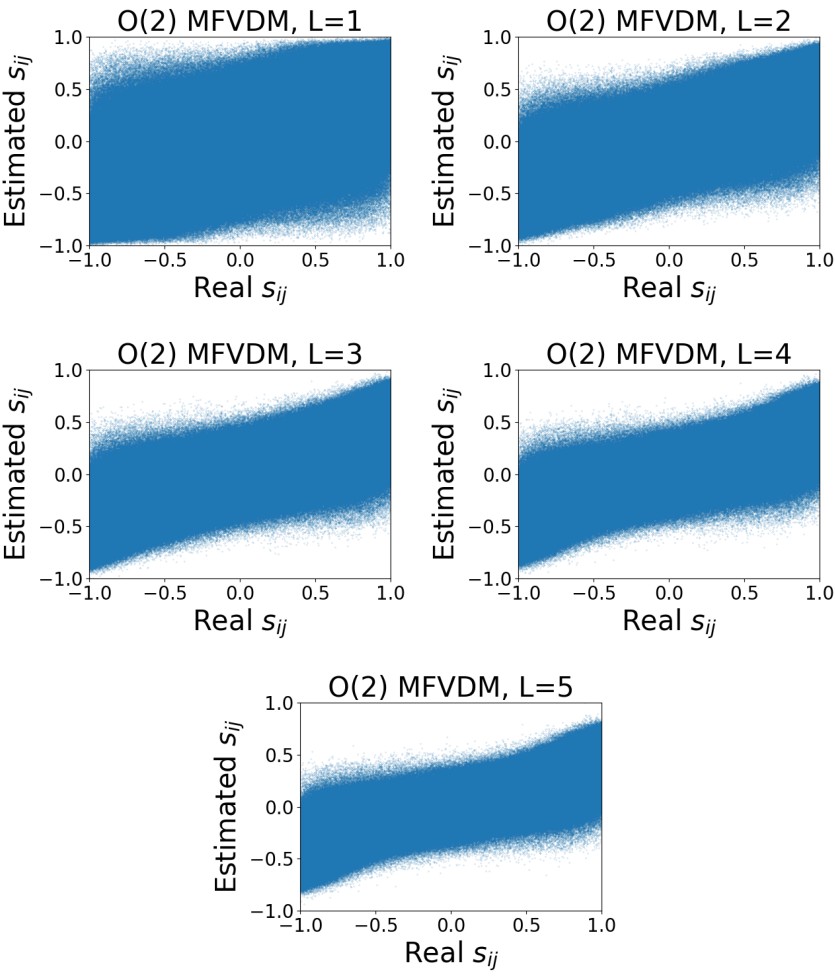

Figure 8: Effect of different choices of maximum frequency $L$ on the estimated cosine similarities. Including higher frequencies improves the estimation. Here, we used 2000 samples and $SNR = 0.25$.

In our experiments, we include our method in the pipeline of Python ASPIRE. Consider a dataset of $N$ images of resolution $D = 97$ pixels. First, the pipeline estimates the amount of noise in the images and performs a first denoising. The resulting images are used to compute a $O(2)$-steerable PCA. The steerable features extracted from each image are used to compute $SO(2)$-invariant vectors via bispectrum [15]. These invariant features are further reduced via a randomized PCA. Then, we perform a $K$-nearest neighbor search on an *augmented* dataset, containing mirrored copies of each image. This provides the connectivity of the synchronization graph, together with an estimation of the relative reflections. Note that the use of an augmented dataset is necessary since the bispectrum computed is only $SO(2)$ invariant; this augmentation is only needed to compute the edges associated with a reflection but does not result in a larger number of nodes in the graph. For each edge retrieved, the optimal rotation is found efficiently by combining a Polar transform and a Fast Fourier Transform. We refer to ASPIRE's documentation for more details.

Next, the relative poses found can be denoised by using the MFVDM methods, i.e. with $SO(2)$ or $O(2)$ and with different values of the maximum frequency $L$. Optionally, a subset of $n < N$ images can be selected and denoised by averaging their nearest neighbors.

We consider the common line synchronization (CL) [7, 23, 24] as a baseline to compare our VDM-based method. Once the poses are estimated, we perform a first 3D reconstruction by using the variational method *Version 2* described in Apx. F.3. We first fit a Gaussian posterior on the density by using the poses estimated in the previous phase and, in a second step, we fine tune both the posterior and the poses. We use the noise variance estimated by ASPIRE from the data to define the generative process of the render and the variance of the images (corrected by the noise variance) to define a Gaussian prior on the 3D density. In particular, the first step uses a batch size of 32, a learning rate of $1e - 2$ and 500 iterations. The second fine-tuning step, instead, uses a batch size of 96, a learning rate of $1e - 3$ and 500 iterations. Finally, we use this low-resolution reconstruction as an initialization for the `3D Refine` method in RELION.

### F.3 Additional details about the *initial reconstruction* method in Sec. 5.2 and other experiments on simulated Cryo-EM dataset

In this section, we provide more details about the method used for *initial reconstruction* in Sec. 5.2 and some additional experiments with variations of it.

In our experiments, we followed an approach similar to [31] based on variational inference to perform the 3D density estimation. The method is based on an differentiable render - based on the Fourier Slice Theorem - which allows to generate projections from a 3D density and back-propagate the errors of the rendered images to the parameters of the density. Like [31], we ignore the contrast transfer function (CTF) in the rendering step for simplicity.

We describe the three versions we implemented in the following three paragraphs. In Sec. 5.2, we used the *Version 2*, initialized with the poses provided by the previous synchronization stage. In Apx. F.3.1, we also experimented with *Version 2* and *Version 3*, combined with different pose initializations.

**Version 1: 3D density estimation** First, we review the basic version of the method, which only performs reconstruction and assumes the poses to be known. As in on [31], we optimize the mean and the variance parameters of a Gaussian distribution over the 3D density. While [31] parameterize the distribution in the Fourier domain (a mean and a variance parameter per frequency), we do so in the image domain. This enables us to leverage a Gaussian prior in the image domain and enforce the 3D density to be zero beyond a diameter of $60\%$ of the image's resolution. The parameters are estimated by minimizing an upper bound on a KL divergence between the estimated and the real posterior distributions on the raw images via stochastic gradient descent (SGD). More precisely, at each iteration, we sample a batch of images and their respective poses, we sample a 3D density from the parameterized Gaussian distribution and project it according to each image's pose. Assuming a fixed Gaussian noise on the images, we compute a KL divergence between Gaussian distributions centered at each image and the projection. The loss also includes the divergence between the 3D density's learnt distribution and a Gaussian prior. Finally, we minimize this loss by back-propagating it on the density's parameters and performing gradient descent. We estimate the noise variance in the images with ASPIRE by looking at the raw images' statistics in their outer rings. Similarly, we

Table 4: Pose Correlation (and Negative Log-Likelihood) on test images with different poses initialization.

| n. samples | 1000 | 1000 | 1000 | 3000 | 3000 | 8000 | 8000 |
| SNR | clean | 2.7 | 0.67 | 0.67 | 0.3 | 0.09 | 0.02 |
|---|---|---|---|---|---|---|---|
| Gumbel-Softmax | | | | | | 0.01 (35.3) | 0.01 (50.9) |
| MFVDM-SO(2) | 34.4 (23.6) | 33.9 (33.8) | 33.1 (50.3) | 33.1 (46.4) | 25.2 (27.8) | 18.4 (44.0) | 18.0 (45.0) |
| MFVDM-O(2) | 100.0 (2.7) | 99.6 (10.3) | 99.3 (33.7) | 99.6 (11.0) | 99.0 (27.8) | 88.8 (52.6) | 57.7 (74.4) |
| Ground Truth | — (2.0) | — (10.4) | — (28.7) | — (11.7) | — (20.9) | — (30.2) | — (55.8) |

estimate the prior variance by looking at the raw images' statistics in the central area (and correcting by the noise variance). We keep the mean of the estimated Gaussian distribution as reconstruction.

**Version 2: Fine-tuning the poses**  Moreover, since the renderer is also differentiable with respect to the images' poses, we also adapted it to either estimate or fine-tune the poses. We parameterize poses in $SO(3)$ by using *quaternions* to guarantee a continuous parameterization. In Apx. F.3.1 and Tab. 4, we experiment with three different initializations: the Ground-Truth or the output of MFVDM-SO(2) or MFVDM-O(2).

**Version 3: Pose Marginalization via Gumbel-Softmax**  Conversely, in the Gumbel-Softmax method, for each image, we learn a generic distribution over a finite subset $\mathcal{G} \subset SO(3)$ of the poses. In practice, for each image, we would like to store the log-likelihood of each element in $\mathcal{G}$; this vector should be turned into a probability by using the *softmax* function. At each iteration, we sample a pose $g_i \in \mathcal{G}$ for each image according to its own distribution and use that to project the 3D density. Unfortunately, because this sampling step is not differentiable, we can not directly back-propagate the gradient to the log-likelihood vectors. Hence, instead of naively applying softmax, we sample $g_i \in \mathcal{G}$ by using the Gumbel-Softmax [33], which offers a *reparameterization trick* to compute the gradient of the parameters of a categorical distribution. Additionally, because the discrete set $\mathcal{G}$ loses all geometrical information about the similarity between poses, we diffuse the log-likelihoods over neighboring points in $\mathcal{G}$ before using the Gumbel-Softmax by using a Gaussian kernel defined in the quaternion space. The final poses are estimated by taking the element $g_i \in \mathcal{G}$ with highest log-likelihood. The set $\mathcal{G}$ is found by optimizing the location of $|\mathcal{G}|$ in $SO(3)$ via gradient descent to minimize a potential energy.

### F.3.1   Other experiments on simulated Cryo-EM dataset

Here, we include a few additional experiments with the models described above. In particular, we compare three different initializations of the poses for *Version 2* - with the ground truth, with the estimation produced by our method and with estimations produced via $SO(2)$ alignment - and the Gumbel-Softmax based method *Version 3* In the $SO(2)$ alignment case, we use the similarity metric recovered by MFVDM to build a diffusion operator on $\mathcal{S}^2$, whose top eigenvectors estimate $z_i$; we initialize $(x_i, y_i)$ to random orthogonal frames.

To evaluate the reconstruction, we compute the likelihood of a number of new *clean* images. We estimate the test images' poses by, first, estimating the optimal alignment $g \in O(3)$ between the training images' ground truth and estimated poses and, then, correcting the test images' poses by $g$. We also fine-tune these estimate poses before computing their final likelihood.

To evaluate the methods, we notice that the probabilistic approach also allows us to estimate the quality of a reconstruction in terms of the likelihood of a set of new images.

**Evaluating the reconstruction**  To evaluate a reconstruction, we compute the negative log-likelihood of a number of *clean images* given the estimated 3D density. Since we assume Gaussian noise, this loss is essentially a MSE between the test images and the density's projections. To estimate a test image's pose, we need to estimate the density's rotation with respect to the ground truth poses used to generate the test data. To find the element $g \in SO(3)$ which aligns the 3D density with the test images, we search for the element $g$ which aligns the estimated training images' poses and their ground truth. Then, $g^{-1}$ is close to the optimal alignment of the test images' ground truth poses to the current 3D density. Once this correction is performed, we also fine tune the test poses via gradient descent as in *Version 2*, while keeping the 3D density fixed. Finally, once the test poses are estimated, we report the negative log-likelihood computed with respect to the density's projections.

**Experiments details and parameters** For each method and dataset, we perform a small search over the hyper-parameters by looking at the negative log-likelihood of the training images. The reconstruction with the smallest loss is then evaluated on the clean test images. We use ADAM to optimize the reconstruction parameters and the poses. We use images of resolution $129 \times 129$, a batch size of 8, learning rates in the range $10^{-3} - 5 \cdot 10^{-1}$ to optimize the density and 1000 to 6000 iterations. In the *Version 2* methods, we used a learning rate in the range $10^{-4} - 5 \cdot 10^{-1}$ to optimize the poses. In the Gumbel-Softmax method, we first used *Version 3* with a learning rate in the range $10^{-4} - 5 \cdot 10^{-1}$ to optimize the distributions over the poses for 1000 to 6000 iterations, followed by a *Version 2* phase of 200 iterations with a learning rate of $10^{-6} - 5 \cdot 10^{-1}$ to refine the maximum-likelihood poses previously found. In the MFVDM methods, we use a maximum frequency $L = 6$ and pick the $0.1 - 15\%$ nearest neighbors of each node.

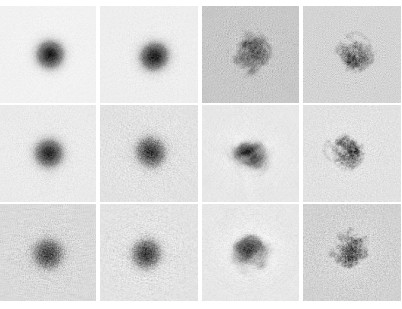

Figure 9: Random projections of reconstructions. *Rows*: 3000, 8000 and 8000 training samples with SNR $0.3, 0.09$ and $0.02$. *Columns*: Gumbel-Softmax, MFVDM-SO(2), MFVDM-O(2) and GT methods.

Table 4 reports the correlation of the training images' poses and the negative log-likelihood (NLL) of 500 clean test images in different training settings. Our MFVDM-O(2) method can accurately recover the poses, yielding reconstruction quality close to the one obtained using ground truth poses (GT). Conversely, we do not find initializing the poses with the spherical embeddings recovered by MFVDM-SO(2) useful. Unfortunately, we find that the NLL - essentially, a mean-squared error - does not reflect the reconstruction quality in the lowest SNR regimes (e.g., the baselines have losses lower than the GT model). Still, the sample projections of the produced densities in Fig. 9 demonstrate that our method (MFVDM-O(2)) achieves similar visual quality to the ground-truth. Instead, the baseline models did not converge to meaningful structures, emphasizing the importance of pose initialization.