# OpenReview forum: "On the symmetries of the synchronization problem in Cryo-EM: Multi-Frequency Vector Diffusion Maps on the Projective Plane"
_NeurIPS.cc/2022/Conference — NeurIPS 2022 Accept_

### Official Review · Reviewer_SWAQ · 2022-07-11

**Rating:** 6
**Confidence:** 3
**Soundness:** 3 good
**Presentation:** 3 good
**Contribution:** 2 fair

**Summary:**

The motivation of this work is to improve the 3D reconstruction quality from cryo-EM generated 2D images and builds on a method called MFVDM.
First, the presented method estimates relative poses between pairs of 2D-images using a group synchronization framework. Here, relative pose can be computed when the viewing angles of the pair of images are either very similar or they differ by 180 degrees.
The authors propose to do a robust computation of similarity of viewing angles and the relative pose (parallel transport) with respect to noise using a multi-frequency approach.
Finally, the authors explicitly estimate the pose of individual 2D images which can then be used to estimate the 3D density of the object. Presented ideas have been tested on a synthetic dataset of vectors created by the authors, which has similar symmetries as cryo-EM data would have. Additional testing was performed on another synthetic cryo-EM dataset.

**Questions:**

In the appendix C, sufficiency for the O(2) relative poses is shown. At what line will the proof fail if relative poses belong to SO(2), ie, O(2) minus the flips?

**Limitations:**

The authors have mentioned the limitation of the work:
(1) This work assumes uniform distribution of poses.
(2)This work does not handle symmetries present in the molecule to be scanned.

**Strengths And Weaknesses:**

Strength:

In other ab initio methods, relative pose between a pair of 2D images can only be estimated if their viewing angles are very similar.
In the presented work, in contrast, the authors are able to also estimate the relative pose when the viewing angles differ by 180 degrees.
The positive effect are that (i) the quality of estimated related poses is improved, and that (ii) the number of connected components in the synchronization graph is reduced.
Secondly, in literature, images belonging to similar viewing angles are aggregated. The aggregated images are then used during 3D reconstruction. However, this aggregation leads to a loss in the obtainable resolution. Hence, in the presented work, the authors estimate the pose of each 2D image and can thereby avoid the aggregation step and the raw images can directly be used in the 3D reconstruction task.
Finally, I believe that the presented analysis of the synchronization symmetries could be helpful for any independent approach to improve 3D reconstruction in cryo-EM.

Weaknesses:

The absence on a proof-of-concept on real-world data is regrettable. Showing the validity and effectiveness of the presented ideas would have made a strong case for the utility of the new approach.
According to the submitted work, VDM assumes the viewing angles to be uniformly distributed. However, this assumption does not hold true for real world cryo-EM datasets. Hence, it would be illuminating to see how the effectiveness of the presented approach deteriorates as the viewing angle distribution is diverging from being uniformly distributed (and maybe getting closer and closer to to the configuration commonly used in cryo-EM).

---

> ### Author Response · Authors · 2022-08-02
> **Reply to Reviewer SWAQ**
>
> We agree with the reviewer that validating our method on real-world data is important.
> We are currently working on some proof-of-concept experiments on a real dataset from EMPIAR, which we hope to include in the next version of the manuscript.
>
> Regarding the proof of sufficiency of $O(2)$ and the failure with $SO(2)$ poses, this is related to the top eigenvalues of the discretized diffusion operator.
> First, note that fields of tangent frames on the sphere correspond to frequency-1 real vectors-fields.
> As argued in Apx B (paragraph "A basis for the invariant subspaces of vector fields"), the diffusion of frequency-1 real vector-fields using $SO(2)$ parallel transport decomposes in two independent complex scalar diffusion operators (with frequencies $\pm 1$), each with an independent 3 dimensional complex top eigen-space.
> Conversely, if $O(2)$ parallel transport is used, this diffusion operator does not decompose anymore, since the flip "mixes" these two otherwise independent spaces in one unique 3-dimensional eigenspace containing the frames.
> We now include more comments about the effect of $SO(2)$ relative poses at the end of Apx C.
>
> Finally, we emphasise that our method can in theory handle symmetric molecules, provided the symmetry is known a priori: a symmetry enforces a precise sparsity pattern in the eigenvalues of the diffusion operators.
> While the top 3-dimensional eigenspace is likely to vanish (since each image can be associated with a number of equivalent poses), other eigenspaces (corresponding to higher frequency spin-weighted spherical harmonics) will emerge.
> In these cases, the top eigenvectors can be interpreted as a multimodal function over $SO(3)$ per image; the modes of such function represent equivalent choices of poses for the image.

---

### Official Review · Reviewer_PHyx · 2022-07-11

**Rating:** 5
**Confidence:** 2
**Soundness:** 3 good
**Presentation:** 2 fair
**Contribution:** 3 good

**Summary:**

This papers formally describes symmetries present in SPA cryoEM and proposes a method to enable the Vector Diffusion Map to recover particle poses using the insights from symmetries. The method can potentially replace current 2D classification methods and allow to skip clustering and averaging steps of cryoEM SPA pipeline altogether. The authors validate their approach on one synthetic and one experimental datasets, and show accurate pose estimations.

**Questions:**

1. One of the advantages of the proposed approach is that it would allow to skip 2D classification step, use all images for the final 3D reconstruction and subsequently increase the final resolution. Did you explore whether this holds true? Moreover, are there any estimations performance wise?

**Limitations:**

The authors adequately addressed the limitations and potential negative societal impact of their work. The authors also thoroughly list the limitations of their method in the last paragraph of section 3.2.

**Strengths And Weaknesses:**

Strengths:
1. The paper establishes formal description of the symmetries present in cryoEM.
2. The validation suggest that the proposed method can accurately recover poses.

Weaknesses:
The paper is very careful in its formal description, which simultaneously is a strength and a weakness depending on readers' math well-read-ness. As a cryoEM practitioner, I found it very hard to follow and felt out of my depth (and reflected in my confidence score).

---

> ### Author Response · Authors · 2022-08-02
> **Reply to Reviewer PHyx**
>
> We thank the reviewer the comment: we will try to add a more intuitive description about our method in the next version to make it more accessible to larger audience.
>
> We agree that it is important to very empirically our initial hypothesis that skipping averaging improves the final resolution.
> To do this, we consider a new baseline and integrate our pipeline into RELION to a more realistic comparison.
> The baseline uses 2D classification and averaging to generate a smaller denoised set of images, and their poses are estimated with a popular common-lines based synchronization method [1, 2] implemented in Python ASPIRE.
> We evaluate both methods by performing an initial reconstruction using the gradient-based variational method from [3] and, then, using RELION to refine it and estimate the final resolution (using the standard Fourier Shell Coefficient metric).
> See Apx E.2.
>
>
> [1] Yoel Shkolnisky and Amit Singer. Viewing direction estimation in cryo-em using synchroniza- tion. SIAM journal on imaging sciences, 2012
>
> [2] Amit Singer, Ronald R. Coifman, Fred J. Sigworth, David W. Chester, and Yoel Shkolnisky. Detecting consistent common lines in cryo-em by voting. Journal of Structural Biology, 2010.
>
> [3] Karen Ullrich, Rianne van den Berg, Marcus Brubaker, David Fleet, and Max Welling. Differentiable probabilistic models of scientific imaging with the fourier slice theorem, UAI (2019)

---

### Official Review · Reviewer_XAUC · 2022-07-14

**Rating:** 6
**Confidence:** 3
**Soundness:** 3 good
**Presentation:** 3 good
**Contribution:** 3 good

**Summary:**

This paper studies the symmetries in group synchronization for pose estimation in cryo-EM. The authors find that relative poses between images in O(2), but not SO(2), provide sufficient constraints to identify image poses in SO(3). This insight is used to improve the multi-frequency vector diffusion map algorithm (MFVDM), extending its use from 2D classification of images to absolute pose estimation that can be used for ab initio 3D reconstruction. The authors show results on a synthetic cryo-EM dataset with varying noise levels.

**Questions:**

See above, what is the computational complexity (time and memory) of the algorithm?
Can the authors report the error between the estimated and ground truth poses (e.g. mean/median error in degrees) instead of correlation?
Please include more detail on dataset generation and the experiments. In the supplement, the authors mention that poses are also updated from their initial values by backprop. What are the details here (e.g. the balance between pose updates vs. model updates) and how much are poses refined?


**Limitations:**

The authors discuss the method's limitations (assumption of uniformly distributed poses, which typically does not hold in practice and testing on synthetic projection images without CTF). The authors could further discuss the computational limitations of MFVDM.

**Strengths And Weaknesses:**

The paper is well-written and motivated. I did not look deeply into the prior work on vector diffusion maps, but the insight is interesting, and the additional capability of absolute pose estimation is significant.

Some details on the synthetic cryo-EM dataset generation and experiments are missing. More discussion of the algorithm's computational complexity would be interesting to include as well.

---

> ### Author Response · Authors · 2022-08-02
> **Reply to Reviewer XAUC**
>
> We are happy the reviewer found our manuscript well-written and our idea interesting and well-motivated.
>
> The synthetic cryo-EM dataset is generated by using the publicly available implementation of [1]. We now include additional details about the reconstruction method and the way the poses are updated in Appendix E.1.
>
> In the next version of the paper, we will replace the old simpler experimental set up used in the Sec. 5.2, with a more realistic and complete one (by integrating ASPIRE and RELION), as we now describe in Apx. E.2.
> The revised version of the paper just uploaded already includes some new experiments in Apx. E.2, where we also report the rotational errors in radians together with the correlations.
>
> Appendix F now includes an analysis of the computational complexity of our method once integrated in a pipeline similar to ASPIRE.
>
> [1] Karen Ullrich, Rianne van den Berg, Marcus Brubaker, David Fleet, and
> Max Welling. Differentiable probabilistic models of scientific imaging with the
> fourier slice theorem, UAI (2019)

---

### Meta-Review · Area_Chair_UGc2 · 2022-09-01

**Recommendation:** Accept
**Confidence:** Less certain

**Metareview:**

This paper formally studies the symmetries in group synchronization for pose estimation in cryo-EM. The main insight obtained by the authors is that the relative poses between images in O(2) provide sufficient constraints to identify image poses in SO(3). This insights leads to improved quality of the multi-frequency vector diffusion map algorithm for 3D reconstruction. A main weakness of the paper is its lack of a proof-of-concept on real-world data. This would have made a much stronger case for the utility of the proposed approach. The current work uses synthetic cyro-EM datasets, which assume a uniform distribution on the viewing angles. This assumption might not hold true for real-world datasets.

**Award:**

No

---

### Decision · Program_Chairs · 2022-09-14

Accept